# Surface engineering of inorganic solid-state electrolytes via interlayers strategy for developing long-cycling quasi-all-solid-state lithium batteries

Ju-Sik Kim [1,4] ✉, Gabin Yoon [1,4], Sewon Kim [1], Shoichi Sugata[2], Nobuyoshi Yashiro[2], Shinya Suzuki[2], Myung-Jin Lee[1], Ryounghee Kim[1], Michael Badding[3], Zhen Song[3], JaeMyung Chang[3] & Dongmin Im [1] ✉

Lithium metal batteries (LMBs) with inorganic solid-state electrolytes are considered promising secondary battery systems because of their higher energy content than their Li-ion counterpart. However, the LMB performance remains unsatisfactory for commercialization, primarily owing to the inability of the inorganic solid-state electrolytes to hinder lithium dendrite propagation. Here, using an Ag-coated $Li_{6.4}La_3Zr_{1.7}Ta_{0.3}O_{12}$ (LLZTO) inorganic solid electrolyte in combination with a silver-carbon interlayer, we demonstrate the production of stable interfacially engineered lab-scale LMBs. Via experimental measurements and computational modelling, we prove that the interlayers strategy effectively regulates lithium stripping/plating and prevents dendrite penetration in the solid-state electrolyte pellet. By coupling the surface-engineered LLZTO with a lithium metal negative electrode, a high-voltage positive electrode with an ionic liquid-based liquid electrolyte solution in pouch cell configuration, we report 800 cycles at 1.6 mA/cm² and 25 °C without applying external pressure. This cell enables an initial discharge capacity of about 3 mAh/cm² and a discharge capacity retention of about 85%.

As carbon neutrality is becoming a pressing issue for sustainability, the development of lithium-ion batteries (LIBs), a key technology for electric vehicles and electric storage systems for smart grids, has attracted significant interest. To push the limits of LIBs, all-solid-state Li batteries have recently drawn considerable attention owing to their high energy density (~900 Wh/L) and safety[1–4]. In particular, the use of Li metal as an anode can considerably increase energy density; the non-flammable feature of inorganic solid electrolytes can prevent potential fire hazards caused by thermal shock or short-circuit, even in large-format cells[2].

Garnet-structured oxide solid electrolytes ($Li_7La_3Zr_2O_{12}$, LLZO archetype) have been widely used owing to their high ionic conductivity (~1 mS/cm @ 25 °C) and good reduction stability to Li metal[5–10]. However, there are plenty of obstacles preventing its practical implementation, including dendrite penetration through solid electrolytes[11–15]. Several approaches, such as coating organic/inorganic layers to protect interface and increase the wettability, modifying the interfacial nanostructure to reduce the local current density, and chemical treatment to remove native layers at the electrolyte surface, have been attempted to alleviate this issue[16–20]. Efforts to enhance Li

---

[1]Battery Material Lab., Samsung Advanced Institute of Technology, 130, Samsung-ro, Yeongtong-gu, Suwon-si, Gyeonggi-do 16678, Republic of Korea. [2]Samsung R&D Institute Japan, Samsung Electronics, 2-1-11, Semba Nishi, Minoh, Osaka, Japan. [3]Sullivan Park Campus, Corning Incorporated, 21 Lynn Morse Rd, Painted Post, NY 14870, USA. [4]These authors contributed equally: Ju-Sik Kim, Gabin Yoon. ✉e-mail: jusik.kim@samsung.com; dongmin.im@samsung.com

wetting using molten Li or metallic coating materials could effectively reduce the initial interfacial resistance, but the long-term cycle performance of solid-state Li metal cells was not satisfactory, and the critical current density was predominantly under 1.5 mA/cm², which is lower than the practical operating condition[18–20]. High-porosity dense, layered garnet-type structures are reported to hinder Li dendrites formation in symmetric Li‖Li cell configuration under a high current density (~10 mA/cm²) by increasing the interface area, but the effect was diminished when the Li metal negative electrode is coupled with an S-based positive electrode[21]. It was also demonstrated that the acid treatment on garnet considerably increased the stability against Li metal dendrites by simultaneously removing the native oxide layer and increasing the interfacial contact area, but the long-term cycling was demonstrated only at elevated temperatures (60 °C)[22].

Introducing a carbon-based interlayer between the inorganic solid electrolyte and the negative electrode has recently been proven effective at suppressing dendrite propagation. Using a Ag-C composite interlayer and sulfide solid electrolyte, a reversibility for ~1000 cycles was demonstrated in a pouch cell at 60 °C with a high current density of 3.4 mA/cm² and an external pressure of 2 MPa[23]. It was demonstrated that a crucial role of the interlayer is to guide the Li deposition/ stripping occurring between the interlayer and current collector. Consequently, the direct contact between Li metal deposit and the solid electrolyte, which commonly leads to non-uniform current distribution, and the resulting penetration of Li dendrites through the solid electrolyte, can be effectively prevented during charge/discharge cycles. However, the mechanism of the Li deposition site regulation remains unclear, because the mixed ionic-electronic conducting nature of the Ag-C composite would promote Li deposition/stripping at the interface between the solid electrolyte and Ag-C composite.

In this study, we first explored the thermodynamic origin of preferable Li deposition toward current collector in the presence of carbon-based interlayer. We then incorporated a Ag-C composite interlayer onto Ta-doped LLZO (LLZTO) with a nanometer-thick Ag layer sputtered on the LLZTO surface. This solid-state electrolyte/

interlayer strategy is applied at pouch cell level using a Li metal negative electrode, a high-voltage positive electrode and an ionic liquid-based liquid electrolyte solution. The single-layer lab-scale pouch cell is efficiently cycled for 800 cycles at 1.6 mA/cm² and 25 °C without applying external pressure delivering an initial discharge capacity of about 3 mAh/cm² and showing a discharge capacity retention of about 85%.

## Results

### Role of interlayer to regulate Li deposition site

The key role of the interlayer is to induce Li plating away from the solid electrolyte, preventing direct contact between Li metal and the solid electrolyte and suppressing dendrite penetration. Recent studies using sulfide electrolyte suggested that carbon-based layers serve this purpose, but a fundamental understanding of this mechanism remains lacking[23]. Therefore, before implementing carbon-based composite onto LLZTO, we attempted to elucidate (via modelling and experimental measurements) the Li deposition behavior in the presence of interlayer and determine whether carbon-based interlayer could regulate the Li deposition site in contact with LLZTO.

Figure 1a, b illustrates two possible scenarios of Li deposition during charging. Li can either be deposited between the solid electrolyte and interlayer, or between the interlayer and current collector. Without an additional driving force, Li⁺ transported from the solid electrolyte should be reduced and plated between the solid electrolyte and interlayer (Fig. 1a) because of the electron-ion mixed-conducting nature of the carbon-based interlayer. However, to the contrary, it was observed that Li deposition occurs at the interface between the current collector and interlayer (Fig. 1b)[23]. This indicates that the reduced Li moved through the interlayer and is plated near the current collector. To understand the driving force behind this Li transport, we conducted DFT calculation to assess the thermodynamically favorable site for Li deposition.

As shown in Fig. 1a, b, the internal arrangement of cell components after charging varies with the Li plating site, leading to a distinct

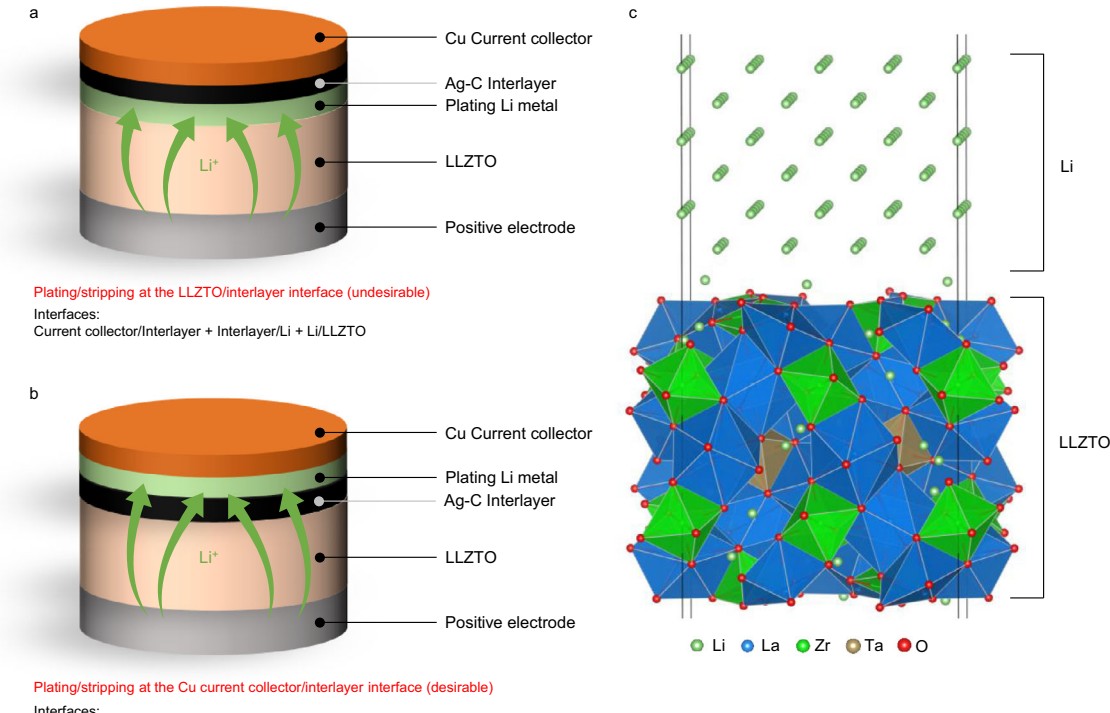

**Fig. 1 | Two possible scenarios of Li deposition in LMBs with interlayer. a** An undesirable case where Li plates/strips at the LLZTO/interlayer interface. **b** A desirable case where Li plates/strips at the Cu current collector/interlayer interface. **c** Atomic model of LLZTO/Li interface.

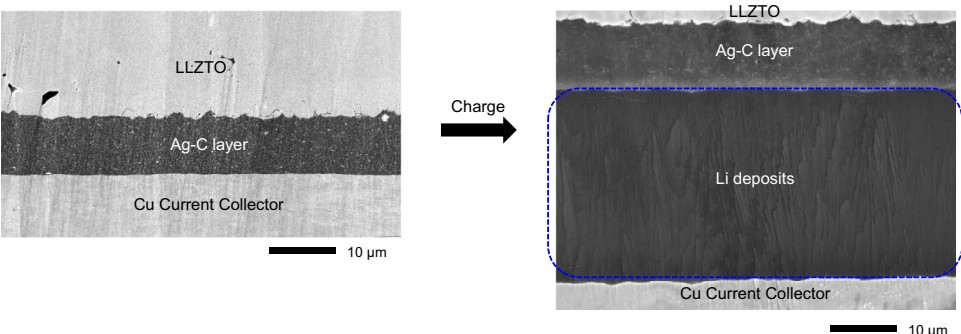

**Fig. 2 | Li deposition behavior in a lab-scale cell with the Ag-C composite attached to a LLZTO pellet.** Ex situ cross-section SEM image of the cell before and after lithiation, where current collector side plating is distinctly observed.

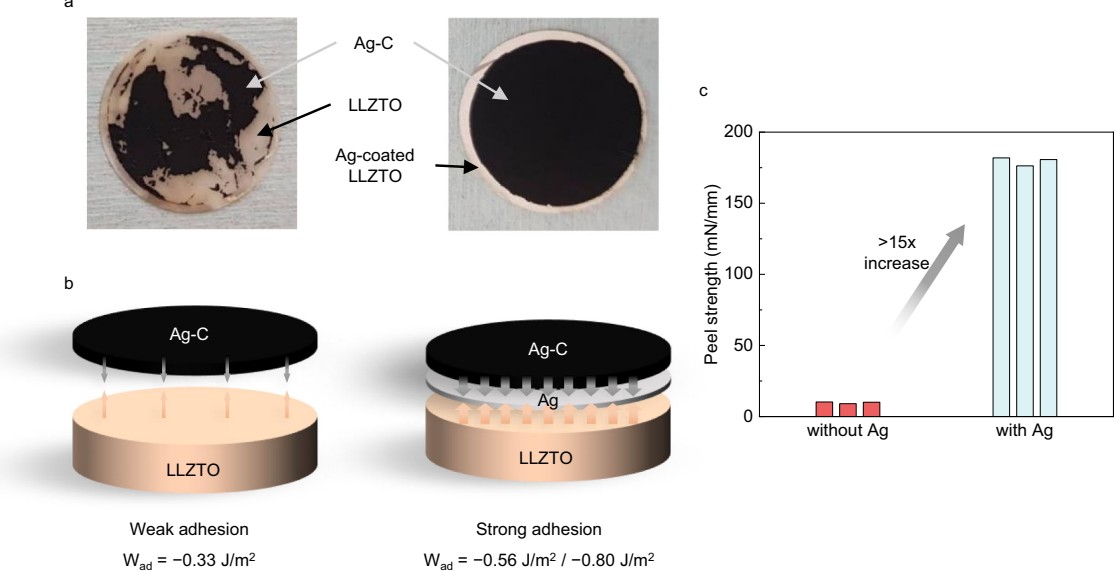

**Fig. 3 | Improvements of adhesion between LLZTO and Ag-C interlayer by introducing Ag coating layer. a** Optical image of LLZTO after the transfer of Ag-C interlayer (left) without and (right) with Ag coating layer. **b** Schematic models of LLZTO/interlayer interface (left) without and (right) with Ag layer, illustrating the role of Ag as a binding agent. **c** Measured peel strength between LLZTO and Ag-C interlayer. Note that 3 bars in the graph refer to three different measurements for the same material.

interface for each configuration. For example, when LLZTO is used as the solid electrolyte and Cu as the current collector along with graphitic carbon (C) interlayer, two Li deposition sites exist: LLZTO/C interface and C/Cu interface. If Li is plated between LLZTO and the interlayer (Fig. 1a), LLZTO/Li, Li/C, and C/Cu function as the interfaces. On the other hand, if Li is plated between the interlayer and current collector (Fig. 1b), the interfaces in the cell are LLZTO/C, Li/C, and Li/Cu. Therefore, by comparing total interface energies, we could predict the thermodynamically favorable Li deposition site. To assess the interface energies, atomic models were built for each interface, as shown in Fig. 1c, for the LLZTO/Li interface, and the work of adhesion ($W_{ad}$) was calculated accordingly (see "Methods" section for interface modeling process). Supplementary Table 2 summarizes the calculated adhesion energies of various interfaces. From these adhesion energies, we could compare the sums of the interfacial adhesion energies between the scenarios and predict the preferable Li deposition site (Supplementary Table 3). For example, in the scenario of graphite interlayer and Cu or Fe, which represents stainless steel (SUS) current collector, our density functional theory (DFT) calculations suggest that the latter case (Fig. 1b) is the more favorable state by -0.51 J/m$^2$ or -0.98 J/m$^2$, respectively. Notably, using the Li$_6$PS$_5$Cl solid electrolyte results in an identical trend (favorable deposition at the Cu current collector side by -0.2 J/m$^2$), corroborating the recent experimental

observation[23]. This indicates that the thermodynamically favorable Li deposition site is between the current collector and interlayer, unless there is a kinetic barrier for Li transport through the interlayer. During charging, Li$^+$ transported from the solid electrolyte would be reduced at the interface between the solid electrolyte and mixed ion–electron conducting interlayer. If the kinetics of Li transport through interlayer is sluggish compared to the charge rate, Li would be undesirably plated at LLZTO/C interface, although it is thermodynamically unfavorable. In other words, Li transport through the interlayer should be faster than Li influx from the cathode side, which is dependent on the charge rate. In this regard, carbon-based interlayer could aid Li transport. According to our nudged elastic band (NEB) calculations, activation barrier of Li-ion diffusion along the surface of carbon is as low as 0.25 eV (Supplementary Fig. 3), which suggests the possibility of fast Li diffusion through the carbon-based interlayer. Therefore, it is crucial to enhance Li transport kinetics through the interlayer for fast charge/discharge. Notably, the above discussion is valid for the ideal interfacial contact; any imperfections such as defects or impurities that may occur in real cases could lead to different outcomes by changing the adhesion energy ($W_{ad}$) of interfaces.

The computational analyses discussed above demonstrate that the use of carbon-based interlayer could protect the LLZTO/Li interface. Encouraged by this understanding, we attempted to employ a

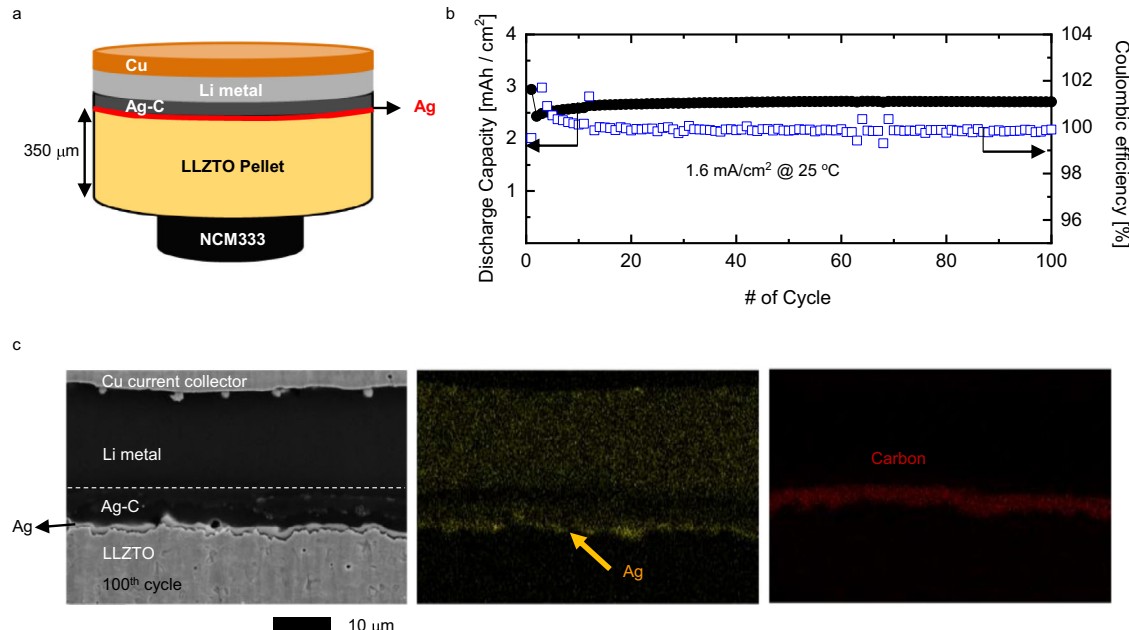

**Fig. 4 | Electrochemical performance of Li|Ag-C/Ag/LLZTO/IL|NCM333 (3.2 mAh/cm² ) single-layer pouch cell. a** Schematic of assembled cell using LLZTO pellet. A relatively small cathode (0.4 cm in diameter) was used to eliminate the possibility of direct contact between the ionic liquid and lithium metal for the cell. **b** Cycling performance of Li|Ag-C/Ag/LLZTO/IL|NCM333 (3.2 mAh/cm²) single-layer pouch cell measured at 25 °C under a 1.6 mA/cm² current density and a charging voltage cutoff of 4.3 V (vs. Li/Li⁺). **c** Cross-section SEM image and elemental analysis of the cell (**b**) after 100 cycles.

Ag-C interlayer onto LLZTO surface to prevent direct contact between Li and LLZTO. For proof-of-concept, we fabricated a lab-scale cell with the Ag-C composite attached to a thick LLZTO pellet (~350-μm-thick) and examined the Li deposition site after charging (see "Methods" section for the fabrication process). Ex situ postmortem scanning electron microscopy (SEM) measurements and analyses (Fig. 2) show that Li penetrates the Ag-C interlayer and is plated between the Ag-C interlayer and current collector, indicating that the interlayer strategy could be successfully employed with garnet-type oxide solid electrolytes.

## Integration of the Ag-C composite onto LLZTO

In the previous section, we demonstrated that the Ag-C interlayer can effectively regulate the Li deposition site. Moreover, to realize competitive energy density in a cell level, we attempted to integrate a Ag-C composite onto thin (<100 μm) tape-cast LLZTO. However, we observed that the clean transfer of the Ag-C interlayer onto ~100-μm-thick tape-cast LLZTO was challenging in some cases, and the transfer process lacks reproducibility. As shown in Fig. 3a, a portion of Ag-C (portion of the electrode in black) was often not attached to LLZTO (portion of the electrode in light pink), resulting in a delaminated and inhomogeneous surface. Even after cold-isostatic pressing (CIP), we frequently observed the partial detachment of the interlayer. We observed that the adhesion between Ag-C composite and LLZTO is weak. DFT calculation results demonstrate that the adhesion energy ($W_{ad}$) between LLZTO and carbon is −0.33 J/m², which is lower than that of the LLZTO/Li interface (−0.71 J/m²). Applying significantly higher pressure could be an effective approach for enhancing the adhesion of ductile sulfide solid electrolytes or robust thick oxide pellets, where the solid electrolyte and Ag-C can withstand high stress. However, thin LLZTO tape is highly brittle; therefore, it is imperative to determine an alternative method to enhance the adhesion between LLZTO and AgC composite.

To achieve a well-attached interface between the interlayer and solid electrolyte, we attempted to introduce an additional layer serving as an adhesive agent at the interface. Additional to binding, this layer requires possessing the following characteristics. First, it should not hinder Li diffusion during charge and discharge, because slow Li diffusion might result in Li plating between the solid electrolyte and interlayer. It should also be chemically and electrochemically stable with LLZTO and the Ag-C interlayer. Considering these criteria, we decided to employ Ag, which was recently demonstrated to facilitate Li nucleation and enable fast Li diffusion in Li−Ag alloys[17,24].

Next, we investigated whether Ag could enhance the adhesion between LLZTO and the Ag-C composite. Our DFT calculations indicated that the adhesion between the interlayer and the solid electrolyte could be enhanced with the Ag layer. For example, $W_{ad}$ between Ag-C composite (modeled by graphite) and LLZTO is −0.33 J/m² (Fig. 3b), but Ag provides a more favorable interface with both LLZTO and Ag-C composite. Calculated work of adhesions between LLZTO and Ag, and that between Ag and Ag-C composite are −0.80 and −0.56 J/m², respectively (Fig. 3b and Supplementary Fig. 4).

To experimentally demonstrate the effect of Ag, we introduced a ~200-nm-thick Ag layer by sputtering onto tape-cast LLZTO. Then, we transferred a Ag-C composite onto the Ag-coated LLZTO. In line with theoretical prediction, we visually observed a uniform transfer of interlayer to LLZTO (Fig. 3a). Compared with the Ag-C transfer to pristine LLZTO tape (Fig. 3a), a sharp difference is shown, indicating that Ag successfully binds LLZTO and Ag-C composite. Enhanced adhesion was further quantified by the peel-off test, as shown in Fig. 3c. Peel strength between LLZTO and Ag-C interlayer is ~10 mN/mm without the Ag layer, but it increases to ~180 mN/mm after Ag layer introduction.

The increased binding strength between Ag and LLZTO is attributable to the increased electronic interaction. To examine charge distribution behavior at the interface, we performed differential charge density analysis.

$$\rho_{\text{diff}} = \rho_{\text{interface}} - \rho_{\text{substrate}} - \rho_{\text{film}}, \qquad (1)$$

where $\rho_{\text{diff}}$ is the redistributed charge density upon the formation of interface, while $\rho_{\text{interface}}$, $\rho_{\text{substrate}}$, and $\rho_{\text{film}}$ are the charge densities of the interface, substrate, and film structures, respectively. According to $\rho_{\text{diff}}$ at LLZTO/Ag and LLZTO/Ag-C interfaces, a larger amount of

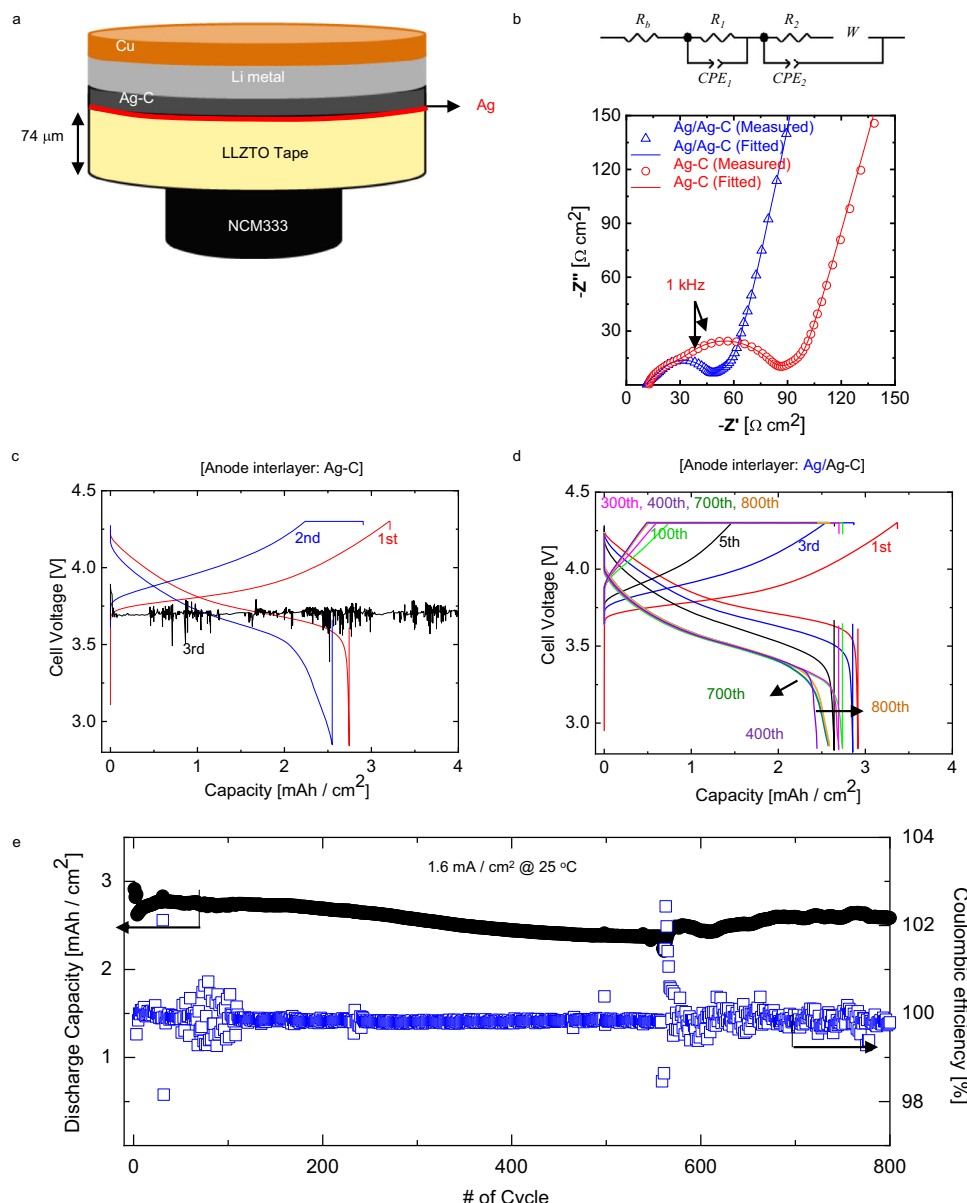

**Fig. 5 | Interface stabilization effect using Ag interlayer in Li|Ag-C/Ag/LLZTO/IL| NCM333 single-layer pouch cells. a** Schematic picture of the assembled cell using LLZTO tape. **b** Nyquist plots of AC-impedance spectra obtained from the Li|Ag-C/ Ag/LLZTO/IL|NCM333 single-layer pouch cells (inset: equivalent circuit). Galvanostatic voltage profiles of the Li|Ag-C/Ag/LLZTO/IL|NCM333 single-layer pouch cells with **c** Ag-C and **d** Ag/Ag-C anode interlayer. **e** Cycling performance of Li|Ag-C/Ag/LLZTO/IL|NCM333 (3.2 mAh/cm²) single-layer pouch cell.

charge is transferred at the LLZTO/Ag interface (Supplementary Fig. 5), which may be responsible for the enhanced adhesion. These results indicate that Ag could successfully aid the binding between LLZTO and AgC interlayer owing to the increased electronic interaction.

## Li metal cells assembly and electrochemical energy storage characterization

Having confirmed that the Ag layer can enhance interfacial binding, we fabricated a cell with a Ag layer and Ag-C interlayer to investigate the battery performance. Electrochemical cells were assembled with both a thick LLZTO pellet (~350 μm) and thin LLZTO tape (<100 μm). Figure 4a shows the schematics of cell configuration fabricated with a LLZTO pellet. We also used an ionic liquid (IL)-based electrolyte solution to favour the wetting and reduce interfacial impedance between the solid-state electrolyte and the positive electrode. In addition, Li metal was initially placed at the negative electrode to improve capacity retention. As described in Supplementary Fig. 6, the anode-free (i.e.,

the negative electrode is only the Cu current collector) cell exhibited a low Coulombic efficiency of 56% and a low discharge capacity of less than 2.0 mAh/cm² even after the first cycle of 0.3 mA/cm² operation. Any pre-existing ductile Li could have diffused into the Ag-C interlayer under 250 MPa CIP during cell manufacturing, thereby filling the inherent voids or pores in the interlayer. This could potentially facilitate a percolating Li transport pathway in the Ag-C interlayer and help to maintain the transport pathway and interface contact without the aid of stack pressure during cell operation. As shown in Fig. 4b, no significant capacity degradation and short-circuiting is observed at the 1.6 mA/cm² operation. Notably, no external pressure was applied, and the cell was operated at 25 °C. The ex situ postmortem SEM images and EDS (Fig. 4c) show that Ag remained in the layer form even after 100th cycle at the Ag-C/LLZTO interface, although some Ag particles were observed to be distributed in the Li metal owing to their thermodynamic preference for the formation of Li–Ag alloy during cycling[23,24]. Diffusion of Ag into Li has been reported in literature[23,25]. Researchers

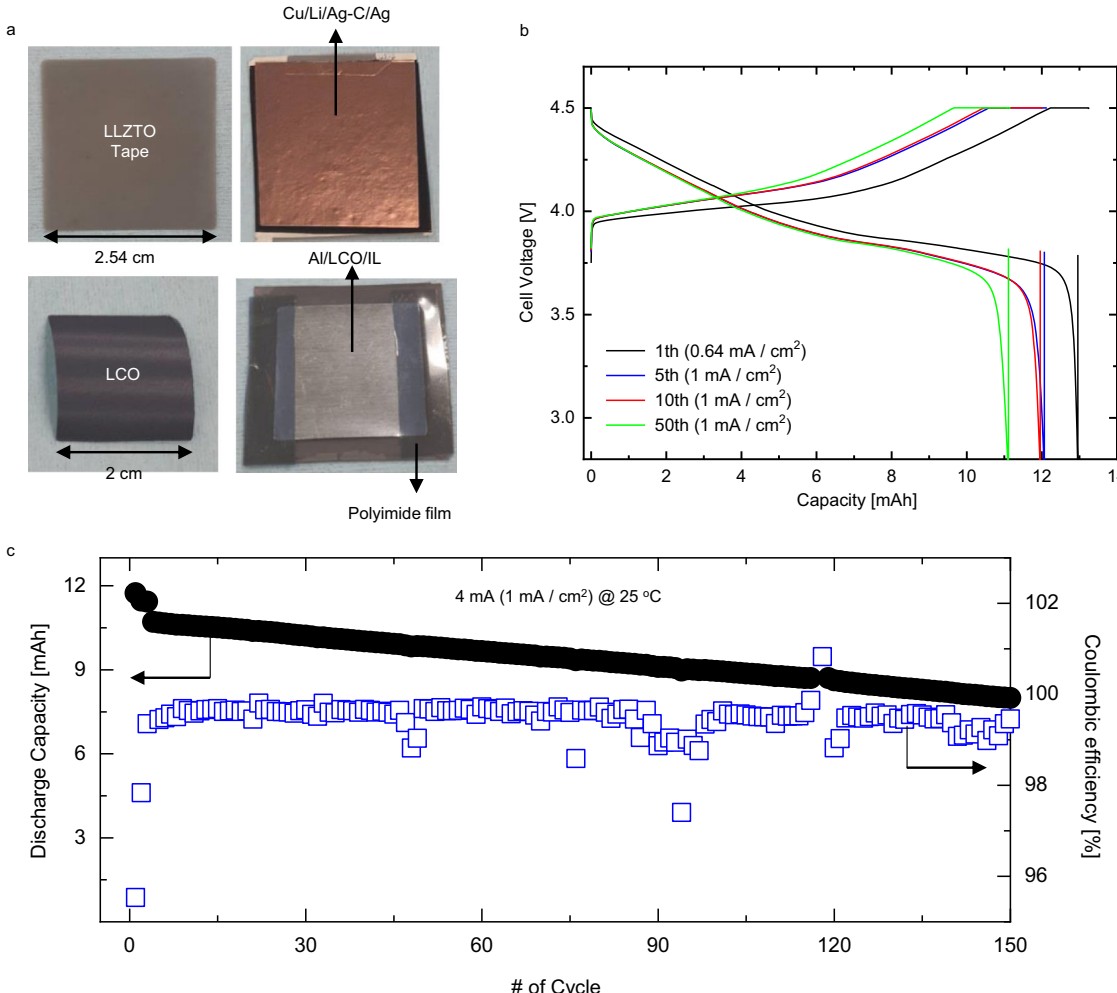

**Fig. 6 | Electrochemical performance of Li|Ag-C/Ag/LLZTO/IL|LCO cell using large-area thin tape solid electrolyte. a** LLZTO tape (2.53 cm × 2.53 cm, 74 mm), Cu/Li/Ag-C/Ag attached to LLZTO tape, LCO-based positive electrode (2 cm × 2 cm) and the assembled Li|Ag-C/Ag/LLZTO/IL|LCO cell. **b** Galvanostatic voltage profiles of the Li|Ag-C/Ag/LLZTO/IL|LCO cell with Ag/Ag-C anode interlayer. **c** Cycling performance of Li|Ag-C/Ag/LLZTO/IL|LCO single-layer pouch cell at 25 °C without external pressure.

attributed this phenomenon to the chemical potential change due to Li–Ag alloying[25]. According to their estimation, the chemical potential change in Ag is expected to be much higher than that in Li. Therefore, the Ag atoms in the coating layer and Ag-C interlayer migrate toward the plated metallic Li during charging, forming Li–Ag alloy phases. As expected from theoretical results, this indicates that reversible Li deposition and stripping occurred between the interlayer and current collector, preventing contact between Li and LLZTO. In addition, it demonstrates that the interface between the anode interlayer and current collector remained intact for long cycles.

To elucidate the effect of Ag interlayer on the electrochemical performance of the thin LLZTO tape, we fabricated a lab-scale cell with cylindrical geometry comprising of 74-µm-thick LLZTO, 3.2 mAh/cm² NCM333-based positive electrode and ionic liquid-based electrolyte additive, as shown in Fig. 5a. Because the predominant issue for utilizing the LLZTO tape was the contact between LLZTO and the Ag-C interlayer (Fig. 3a, b), we carried out electrochemical impedance spectroscopy (EIS) measurements to investigate the change in interfacial resistance. Figure 5b shows that the complex impedance plot presenting the EIS data consists of two depressed arcs, which slightly overlap in the high-frequency region, Warburg impedance, and a capacitive line (blocking region) in the low-frequency range[26–28]. The equivalent circuit, shown in the inset of Fig. 5b, was implemented to quantitatively analyze the experimental impedance spectra. The

Ohmic resistance ($R_b$), overall interfacial resistance ($R_{ct} = R_1 + R_2$), and Warburg resistance ($W_R$) were evaluated using the complex nonlinear least-squares (CNLS) fitting method; the values are listed in Supplementary Table 4. There are no drastic changes in the $R_b$ and $W_R$ for both cells (i.e., LLZTO with and without Ag coating); $R_{ct}$, which is associated with the charge-transfer reactions at the cathode–anode interface, significantly decreases from 66.4 to 32.5 Ω cm² when Ag is present on the LLZTO surface. Assuming that the cathode interfacial kinetics is unchanged in the presence of Ag, the decrease in the $R_{ct}$ is attributable to the improved charge-transfer kinetics at the negative electrode-LLZTO interface due to the increased effective area and the enhanced bonding of anode to LLZTO.

Accordingly, the charging/discharging curve (Fig. 5c, d) shows that the cell without Ag failed at a current density of 1 mA/cm² by short-circuiting, while the cell with the Ag layer was capable of operating at a high current density of 1.6 mA/cm² without short-circuiting. Galvanostatic cycling tests performed on a Li|Ag-C/Ag/LLZTO/Ag/Ag-C|Li symmetric cell with no ionic liquid-based electrolyte additive confirmed the short-circuit tolerance of the Ag/Ag-C interlayers. The symmetric cell with the Ag/Ag-C interlayer cycled at 1.6 mA/cm² without short-circuiting, whereas the cell without the interlayer short-circuited at 0.4 mA/cm², as shown in Supplementary Fig. 7. Moreover, the overpotential gradually increased upon cycling, diminishing the high-voltage constant-current plateau during cell charging in Fig. 5d.

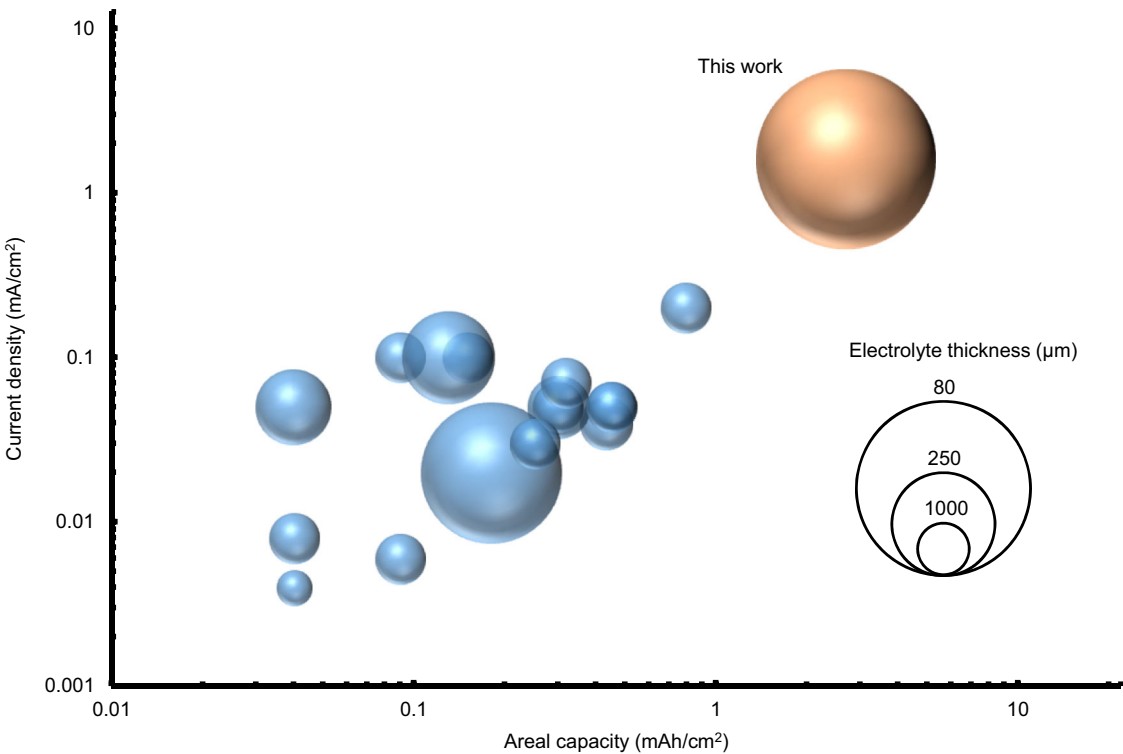

**Fig. 7 | Electrochemical performance of Li|Ag-C/Ag/LLZTO/IL|NCM333 cell comparing with LLZO-based batteries reported so far.** Summary of areal capacities and operating current densities of LLZO-based batteries reported thus far. Full cells, which are reversibly cycled at least 100 times at room temperature, are presented. Additional details such as cell configuration and number of reversible cycles are summarized in Supplementary Table 6.

This might be associated with the physical degradation of the negative electrode/solid electrolyte interface, originating from the local stress due to volume change upon cycling or Ag diffusion into Li metal. Nevertheless, our interlayer could successfully protect LLZTO from dendrite penetration, enabling the delivery of an initial areal discharge capacity of about 3 mAh/cm² and discharge capacity retention of about 85% after 800 cycles at 1.6 mA/cm² and 25 °C for the Li|Ag-C/Ag/LLZTO/IL|NCM333 cell (Fig. 5e). The cycle stability could be also attributed to the chemical stability of LLZTO/Ag interface. In our Ab initio molecular dynamics (AIMD) simulations (Supplementary Fig. 8), it is clearly shown that LLZTO/Li₉Ag₄ interface remains intact for at least 40 ps, which suggests reasonable chemical stability of the interface between LLZTO and Ag/AgC interlayer. Additionally, assuming the Li|Ag-C/Ag/LLZTO/IL|NCM333 cell configuration as a one-unit cell in the pack, we calculated an energy density of about 680 Wh/L (Supplementary Table 5).

We also assembled and tested a Li|Ag-C/Ag/LLZTO/IL|LCO cell using a 3.2 mAh/cm² LCO-based positive electrode and large-area thin tape solid electrolyte (area 2.53 cm × 2.53 cm, thickness 74 μm) under a high charging voltage cutoff of 4.5 V (vs. Li/Li⁺) to demonstrate its applicability in more practical condition. As shown in Fig. 6a, we used a 4 cm² positive electrode to prepare a 12 mAh cell that was tested at 25 °C. The charging–discharging curves of Fig. 6b show that the cell delivers an initial discharge capacity of about 13 mAh (corresponding to ~3.2 mAh/cm² for an electrode area of 4 cm²) at 0.64 mA/cm² under the condition of 4.5 V cutoff. The average discharge voltage of the cell at 1 mA/cm² is calculated to be 3.93 V (vs. Li/Li⁺). Figure 6c shows the long-term cycling performance of the Li|Ag-C/Ag/LLZTO/IL|LCO cell. The cell delivers an initial discharge capacity of about 12 mAh and demonstrates a discharge capacity retention of about 65% after 150 cycles at 4 mA (1 mA/cm²) and 25 °C.

We are confident that the interlayers strategy proposed could be effective for the development of lithium batteries using inorganic solid-state electrolytes. Moreover, the use of the IL-based electrolyte at

the solid-state electrolyte/positive electrode interface enables the operation at 25 °C without applying any external pressure. In particular, for the Li|Ag-C/Ag/LLZTO/IL|NCM333 cell, we report an initial areal discharge capacity of about 3 mAh/cm² and a discharge capacity retention of about 85% after 800 cycles at 1.6 mA/cm² (Fig. 5e). This battery performance expands the state-of-the-art for long-term cycling of lab-scale cell tested at 25 °C employing a solid-state oxide solid electrolyte as summarized in Fig. 7 and Supplementary Table 6[29–44].

## Discussion

Electrochemical reactions in a battery occur at the interface between the electrolyte and the electrodes. Because carbon material is a mixed electronic-ionic conductor, the Ag-C composite interlayer can be considered an electrode. Therefore, if a carbon material is employed as interlayer at the negative electrode, a redox reaction occurs at the interface between the solid electrolyte and interlayer. The Li ions (provided by the positive electrode active material) transported from the electrolyte during charging are expected to be reduced to Li metal at an early stage as carbon possesses limited capability of Li ion storage. If Li stays at solid electrolyte/interlayer interface, the interlayer cannot prevent dendrite penetration owing to the Li growth near the solid electrolyte. However, we have demonstrated that Li can be plated at the interface between the interlayer and current collector. Given that the reduction reaction occurs at the solid electrolyte/interlayer interface, the Li metal is transported from the solid electrolyte/interlayer interface to the interlayer/negative electrode interface. From a thermodynamics perspective, the theoretically calculated interface adhesion energies determine Li stability difference at each interface as a possible driving force for Li transport (Fig. 1a and b). Our analyses identify the following condition to ensure Li deposition on the negative electrode side: The interface adhesion between the current collector and interlayer should be weak (large $W_{ad}$), whereas that between the interlayer and solid electrolyte should be strong (small $W_{ad}$).

It has been widely reported that creep is a major mechanism for Li deformation[45–47] because of the high homologous temperature ($T/T_M$, $T_M$ indicates melting temperature) of Li metal at room temperature (0.66). The continuous Li reduction at the solid electrolyte/interlayer interface during charging leads to consistent stress generation. After Li completely fills the pores and voids in the Ag-C interlayer, this stress drives Li extrusion toward the negative electrode. By this, the solid electrolyte could be protected from dendrite penetration because Li would not grow at the solid electrolyte/interlayer interface. During discharging, the overall process is reversed. The Li metal near the solid electrolyte is first oxidized to Li ion, forming a Li vacancy. Then, the Li metal is deformed to fill the vacancy. During the entire charging–discharging, the solid electrolyte maintains contact with the Ag-C interlayer, preventing abrupt morphology evolution due to the high mechanical stress during Li plating and stripping. Under this mechanism, an increase in the charge current density would result in fast Li accumulation at the solid electrolyte/interlayer interface, leading to severe stress buildup. Thus, two conditions should be satisfied toward warranting cycle stability under high current density: (1) Li transport through the interlayer should be sufficiently fast to rapidly relieve the stress and (2) the adhesion between the solid electrolyte and interlayer should be sufficiently strong to tolerate the local stress. Therefore, a careful design of the interlayer morphology with optimized microstructure, pore structure, and tortuosity would enhance the Li transport kinetics through the interlayer.

In summary, quasi-all-solid-state lithium batteries were prepared using a Ag-coated LLZTO, Ag-C composite interlayer, and NCM333-based positive electrode wetted with an IL-based electrolyte additive. Theoretical calculations demonstrated that an Ag-C interlayer could regulate the Li deposition towards the negative electrode side, preventing the direct contact of Li and LLZTO and effectively suppress dendrite penetration. Sputtered Ag layer was applied to enhance the adhesion between LLZTO and the interlayer. The surface engineering of inorganic solid-state electrolytes via interlayers strategy we propose enabled to build a single-layer lab-scale Li|Ag-C/Ag/LLZTO/IL|NCM333 pouch cell capable of delivering an initial discharge capacity of about 3 mAh/cm$^2$ and showing a discharge capacity retention of about 85% after 800 cycles at 1.6 mA/cm$^2$ and 25 °C without applying external pressure. We also calculated a projected energy density of 680 Wh/L for a Li|AgC/Ag/LLZTO/IL|NCM333 cell as a one-unit cell in the pack. We are confident that the present results provide valuable insights for developing practical LMBs using inorganic solid-state electrolytes.

## Methods

### Materials

LLZTO ($Li_{6.4}La_3Zr_{1.7}Ta_{0.3}O_{12}$) powder was synthesized from a precursor mixture of $Li_2CO_3$ (>99.0%, ChemPoint), $La_2O_3$ (98.6%, MolyCorp), $Ta_2O_5$ (99.99%, Sigma-Aldrich), and $ZrO_2$ (98%, Zircoa Inc.) using a solid-state reaction method. The mixed powder was calcined in air at 950 °C for 5 h followed by 1200 °C for 5 h to yield LLZTO powder. The calcined powder was then ball-milled in air with zirconia balls for 10 min at 300 rpm using a planetary mill (Pulverisette 7, Fritsch, Germany). Ball milling was repeated 12 times at 5-min intervals.

To fabricate a dense pellet, LLZTO powder (100 g) was hot-pressed in a graphite die at 3 kpsi, followed by sintering at 1100 °C for 2 h in Ar atmosphere at a heating rate of 300 °C/h. The relative density of the pellet was estimated to be >98% with respect to the theoretical density of LLZTO calculated from the X-ray diffraction (XRD) data. The pellet was cut into with (14 mm diameter, 360 μm thickness) using a laser cutter in air. Next, it was subjected to ultrasonic cleaning in hexane for 10 min and heat-treated at 800 °C for 1 h in dry air. Finally, the pellet surface was polished to a thickness of ~350 μm using polishing machines (LaboForce-3, Struers).

A thin electrolyte film was prepared via tape casting. LLZTO powder was added to a mixed solvent of toluene and isopropanol and was then mixed for approximately 1 h. Fish oil, polyvinyl butyral, and butyl benzyl phthalate were added to the slurry as a binder and dispersant. The resulting slurry was removed as a thin film on a Mylar sheet, and then dried at 120 °C for 1 h. The solid electrolyte tape was sintered at 1100 °C for 2 h.

The XRD patterns in Supplementary Figure 1 reveal that both pellet and tape have cubic garnet structures. Therefore, there is no significant difference between the material properties of the 350-μm-thick pellet and 74-μm-thick tape solid electrolytes.

Acid treatment was performed for cleaning the LLZTO surface by simply immersing the discs into a glass bottle with 1 M HCl solution (in distilled water) in a dry room (dew point, −60 °C) at 25 °C for 20 min at a weight ratio of 1:10 (pellet:acid solution). To prevent the local variation in concentration in the acid solution due to the released Li or prevent the close contact of the electrolyte to the glass container, the container was rolled in the bottle-roller at approximately 60 rpm during protonation. We then removed the solution, washed the discs with ethanol, and dried them in a dry room.

### Cells assembly and electrochemical measurements

We employed a quasi-all-solid-state cell. In each quasi-all-solid-state cell, an ionic liquid and a solid oxide electrolyte (LLZTO) were used as the cathode and anode electrolytes, respectively. First, a 200-nm-thick Ag layer was coated on the LLZTO surface by radio-frequency sputtering (SNTEK, 16-SN-055) with a Ag target (99.99%). The sputtering was performed for 500 s at 150 W at a working pressure of 5 mTorr under a 40-sccm flow of high-purity Ar gas (99.9999%) at a constant substrate temperature of 22 °C.

An Ag-C layer coated on a 10-μm-thick stainless steel (SUS) foil was prepared using a mixture of carbon black powder (99.7%, average particle size = 38 nm, Asahi carbon) and Ag nanoparticles (D50 = 60 nm)[23]. Ag and carbon black powder were mixed in a weight ratio of 1:3 in N-methylpyrrolidone (Sigma-Aldrich) with 7 wt% of polyvinylidene fluoride (Solvay) as a binder under constant stirring (1000 rpm) for 30 min using a mixer (Thinky Corporation, AR-100). The resulting slurry was then coated on a SUS foil using a screen printer and dried in air at 80 °C for 20 min. The coated Ag-C layer was further dried under vacuum at 100 °C for 12 h. The Ag-C layer was attached as an anode interlayer on the acid-treated LLZTO surface via cold-isostatic pressing (CIP) under 250 MPa. After the SUS foil was peeled, a Li metal foil (99.99%, 20-μm-thick, Honjo Metal Co., Ltd.) was attached to the Ag-C surface via 250 MPa CIP. Commercially available ($Li_{1+x}(Ni_{0.33}Co_{0.33}Mn_{0.33})_{1−x}O_2$ (NCM333, loading capacity: 3.2 g/cc, active material: 96 wt%; thickness: 50 μm; Samsung SDI) and $LiCoO_2$ (LCO, loading capacity: 4.22 g/cc, active material: 97.6 wt%; thickness: 46 μm; Samsung SDI) were employed as a cathode and as a current collector, Al foil (9 μm foil, Nippon Foil Mfg. Co., LTD) was used as received. An N-methyl-N-propyl pyrrolidinium bis(fluorosulfonyl) imide (Pyr13FSI, 99.9%, water content <20 ppm, Kanto Chemical Co. Inc.) ionic liquid mixed with a 2 M lithium bis(fluorosulfonyl)imide (LiFSI, 99.9%, water content <10 ppm) salt was used as the catholyte. The catholyte solution (20 wt% relative to the cathode) was infiltrated into the cathode in a dry room (dew point, −60 °C), followed by maintaining a vacuum state for 2 h. When the residual solution on the cathode surface was removed using Kimwipes, the solution uptake by the cathode was ~7 wt%. We placed the ionic-liquid-infiltrated cathode on the cathode side of the LLZTO in a single-layer pouch cell and then sealed the cell under vacuum (750 Torr).

Potentiostatic electrochemical impedance spectroscopy (PEIS) measurements were conducted at 25 °C with an open circuit in the potentiostatic mode over a frequency range from 0.1 Hz to 10 kHz using AC perturbation of 10 mV with a frequency response analyzer (Solartron, SI 1255 FRA) in conjunction with a potentiostat (Solartron,

SI 1287 ECI). All PEIS data was recorded with 10 frequencies per decade. Before conducting the PEIS measurements, the cells were kept under open-circuit potential for 10 minutes.

A battery cycler (Toscat-3100, Toyo System) was employed to measure the charge–discharge curves of the quasi-all-solid-state cells at 25 °C. The cells were cycled with a constant current (CC)–constant voltage (CV) charging and CV discharging mode in the potential ranges of 2.8–4.3 and 2.8–4.5 V (vs. Li/Li⁺) for NCM333 and LCO, respectively. We evaluated five cells of each sample to ensure data reliability.

Furthermore, the ionic and electronic conductivities of the 350-μm-thick pellet and 74-μm-thick tape solid electrolytes were measured from the symmetric cells (Au | LLZTO|Au) by potentiostatic alternating current (AC) impedance spectroscopy and the Hebb–Wagner polarization method. The electronic conductivities of LLZTO pellet and tape were estimated from the steady-state current under DC polarization with an applied voltage of 0.5 V at 25 °C. For data reliability, we evaluated five symmetric cells for each electrolyte. As shown in Supplementary Table 1 and Supplementary Fig. 2, the pellet and tape electrolytes have almost the same ionic conductivities, electrochemical stability windows up to 6 V vs. Li/Li⁺, and ionic transference numbers ($t_{Li^+}$ = -1).

All the electrochemical test was performed within the environmental chamber of constant temperature (temperature error: ±1 °C, Shin Cooperation).

## Physicochemical characterizations

We examined the cross-sectional microstructure of the interface between the anode and LLZTO electrolytes using an SU-8030 FE-SEM (Hitachi) coupled with an energy-dispersive X-ray spectroscopy (EDS) spectrometer with an accelerating voltage of 5 kV and a working distance of 8 mm. For the sample preparation, the cycled cells were disassembled in an Ar-filled glove box, and after removal of a cathode, the cross-sections of LLZTO in contact with Ag/Ag-C/Li anode were mounted to the sample holder using a load-lock chamber to avoid air exposure.

The adhesion strength between LLZTO and the interlayer was evaluated by measuring the peel strength using a tensile strength tester (AGS-X, Shimadzu). The anode interlayer was attached on the acid-treated surface of the LLZTO by CIP under 250 MPa, and the peel strength was measured by pulling the SUS foil at a cross-head speed of 100 mm/min.

## Computational details

All density functional theory (DFT) calculations were conducted using Vienna Ab initio Simulation Package (VASP)[48]. Exchange-correlation energies were dealt with spin-polarized generalized gradient approximation parameterized by Perdew–Burke–Ernzerhof[49]. We used projector-augmented wave pseudopotentials with a plane-wave basis set as implemented in VASP[50]. To describe van der Waals interaction, DFT-D3 dispersion correction proposed by Grimme et al. was applied, except for metallic structures[51]. A kinetic energy cutoff of 400 eV was used for all calculations involving slab geometry, and structure optimization was performed until the remaining forces converged within 0.02 eV/Å. To model the atomic structure of partially disordered LLZTO, we first generated various disordered structures and performed Ewald summation using *pymatgen* package[52] to screen 30 lowest-energy structures. Here, the Li occupancies at both Li sites (24d, 96 h) were set to the values reported by Awaka et al.[53]. Then, we conducted DFT calculations for the screened structures to determine the lowest-energy structure.

Interface models with sufficiently thick (>15 Å) vacuum slabs were constructed to describe various interfaces in LMBs. To generate interfaces between two different materials, most stable surfaces were first identified for each material. In particular, the (100) surface with a Zr-poor configuration was used for LLZTO following the work of

Canepa et al.[54], and (111) surface was selected for Fe, Cu, and Ag. In addition, (100) and (001) surfaces were used for Li and graphite, respectively. Then, we cleaved bulk structures to form two surface slab models. Here, convergence tests were performed to determine the thicknesses of slabs. Finally, interfaces were formed by aligning two surface slabs. Because of the different lattice parameters of slab models, we allowed slight angle distortion (<3°) or elongation or shrinkage of lattice (<5%) to ensure the lattice compatibility. All interface building processes were performed using the *pymatgen* python package[52].

Structural relaxation of interfaces was conducted in two steps: distance between two surfaces was first optimized; then, one of the surface structures was laterally shifted to determine the most stable alignment between two surface structures. The adhesion energy ($W_{ad}$) was obtained by comparing the energies of interface structure and isolated surface structures.

$$W_{ad} = \frac{1}{A}\left(E_{\text{interface}} - E_{\text{substrate}} - E_{\text{film}}\right), \qquad (2)$$

where $A$ is the area of interface. $E_{\text{interface}}$, $E_{\text{substrate}}$, and $E_{\text{film}}$ are calculated energies of interface, substrate, and film structures, respectively.

Ab initio molecular dynamics (AIMD) simulations were performed to observe the structure evolution of interfaces. Starting from interface models generated by the method described above, we first stabilized the interface by slow heating (0 K to 300 K for 2 ps). After the equilibration, a production run of 40 ps was performed in NVT ensemble at 300 K.

NEB calculations[55] were conducted to evaluate the energy barrier of Li diffusion at the surface of carbon. Seven intermediate images were generated along the diffusion pathways, and structure relaxation was performed to determine the energy profile.

## Reporting summary

Further information on research design is available in the Nature Portfolio Reporting Summary linked to this article.

## Data availability

The data generated or analyzed in this study are available from the corresponding authors upon reasonable request.

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

## Acknowledgements

This work was supported by funds from Samsung Electronics Co. Ltd.

## Author contributions

J.-S.K. and G.Y. conceived and designed the overall experiments, analyzed the data, and wrote the manuscript. J.-S.K. and S.K. performed all electrochemical experiments. G.Y. performed DFT calculations on the adhesion energies of the anode-solid electrolyte interfaces. S.S., N.Y., and S.S. prepared the AgC anode interlayer and measured peel strength between the LLZTO and the AgC interlayer. R.K. examined the cross-sectional microstructure of the interface between the LLZTO and the AgC interlayer. M.B., Z.S., and J.C. prepared the tape-cast LLZTO electrolyte. D.I. supervised the study, discussed the results, and commented on the manuscript. All the authors participated in discussion and provided constructive advice for experimental design.

## Competing interests

The authors declare no competing interests.
