## [Peer Review File · Nature Communications]

REVIEWER COMMENTS

Reviewer #1 (Remarks to the Author):

Reviewer Comment date: 5/16/2022

Ms. No.: NCOMMS-22-14355

Title:

Room-temperature–low-pressure-operating high-energy lithium metal batteries employing garnet-type solid electrolytes and anode interlayers

Authors:

Ju-Sik Kim^{1†*}, Gabin Yoon^{1†}, Sewon Kim¹, Shoichi Sugata², Nobuyoshi Yashiro², Shinya Suzuki², Myung-Jin Lee¹, Ryounghee Kim¹, Michael Badding³, Zhen Song³, JaeMyung Chang³, Dongmin Im

Based on authors report that NCM333//garnet ceramic electrolyte//Ag-coated//Ag-C interlayer//Li//Cu hybrid cell (with IL electrolyte) showed a high energy density of 680 Wh/kg for 800 cycles (coin cell configuration). It seems quite promising. There are some questions need to explain before it can be accepted to publish.

Comment:

1. What is the ionic conductivity of 350 um thick LLZO pellet or 100um thick LLZO tape? Also, the ionic transference number, ESW, etc. It is no information available. It needs to report that basic information.
2. It is confusing for symbol. Ag-coated and Ag-C or AgC in the MS. It needs to distinguish clearly.
3. The cathode material is NCM333, it is better describing it more clearly.
4. In Page 5, IL amount is being infiltrated in cathode. What is the exact amount IL being uptake by cathode?
5. In Fig 1, the model displays look like anode free system without Li metal. But, the authors still applied 20 um Li metal in Full cell. It needs to explain what is the difference?
6. Symmetric cell with IL and without IL exp. needs to explain or prove the tolerance of Li dendritic generation.

7. The authors need to explain why Ag in coating layer and Ag-C interlayer are moving (it is Li-Ag alloy is being formed during the charge). How come? It needs to compare the previous paper results, the Ag is moving to Cu current collector.

Ref: "High-energy long-cycling all-solid-state lithium metal batteries enabled by silver-carbon composite anodes", Nat Energy, <https://doi.org/10.1038/s41560-020-0575-z>

8. In Fig. 4, The I-V curves are needed to support the long-term cycling performance?

9. In Fig. 5, the EIS spectra need a suitable EIS Circuit Model, and please compare Rb, Rct, W parameters in detail.

10. It seems that the I-V curve results are not consistency with the cycling test results. The constant current section is quite short after 100 cycles. Why? Fig. 6(b) is using capacity of mAh/cm², but in Fig. 6(c), the capacity is using mAh?

11. There are a lot of syntax errors; like 12-mA-capacity cell in Page 14, line 380, and 3.4 mA/cm in page 3, line 64, etc.

12. Please describe it more detail, How you get ED of 680 Wh/L?

In summary, I suggest the MS needs to major revision before it responses the questions properly.

Reviewer #2 (Remarks to the Author):

In this work, the authors investigated a novel lithium metal -garnet type solid electrolytes (Ta-doped LLZO) Interface where Ag-C composite interlayer inserted in between. The working principle and formation mechanism of the interface/battery have been specifically investigated. Although the experimental data are systematic and the evidence is sufficient, however, I found the presentation of the manuscript quite simple, even a bit poor, the computational modeling details are unclear. The central topic of the article seems to be unclear enough caused by the unreasonable arrangement of the article, which is unfavorable for readers to understand the work. Some arguments are not logical and compelling, and the novelty of the article could not match the high standard of this high-impact journal. Therefore, the article is suggested to be reorganized, and major revisions are required. The following comments might be helpful.

Comments:

☒• Although the author has described the definition of adhesion energy, a table with the energy values for different simulations will be helpful.

☒• On page 6, lines 163-164; author mentioned shifting one of the surfaces to determine stable alignment between two surfaces after relaxation. It is not clear on what basis the stable alignment was decided.

☒• The authors claimed/showed long cycle stability of the interface (800 cycles at room temperature) however there is no computational support for this point. To investigate the interactions between two surfaces, for example between Li and LLZTO, the ab initio molecular dynamics (AIMD) simulations at operating temperature are needed. Did the author think in this direction?

☒• Another point is LLZO is partial disordered material, how did the author build that structure and how Ti atoms doped in Zr sites.

☒• No figures related to the simulations of the LLZTO-Ag-Graphite.

☒• Page 8, lines 221-223 are unclear. Have the simulations with graphite and Cu or Fe are performed? There is no figure shown detailing it. Moreover, the meaning of the numbers in J/m² are not clearly mentioned.

☒• lines 226-254 the authors discussed and concluded about Li transport through the interlayer without any computational/theoretical evidence. Li transport/migration barriers through the Li/LLZO have been reported in the paper (journal of The Electrochemical Society, (2020) 167 100537). The question coming is what is the Li barrier in this scenario?

☒• The adhesion energy and charge calculations did not fully interpret the thermodynamic and the origin of the preferable Li deposition process.

Response Letter

Reviewer #1

Comments:

Based on authors report that NCM333//garnet ceramic electrolyte//Ag-coated//Ag-C interlayer//Li//Cu hybrid cell (with IL electrolyte) showed a high energy density of 680 Wh/kg for 800 cycles (coin cell configuration). It seems quite promising. There are some questions need to explain before it can be accepted to publish.

Author reply:

We appreciate the reviewer for carefully reviewing our manuscript and for their helpful comments. Their comments, followed by our point-by-point responses, are shown below.

1) What is the ionic conductivity of 350 um thick LLZO pellet or 100um thick LLZO tape? Also, the ionic transference number, ESW, etc. It is no information available. It needs to report that basic information.

Author reply:

Following the reviewer's suggestion, we have measured the ionic and electronic conductivities of solid electrolytes from the symmetric cells (Au | LLZTO | Au) by AC-impedance spectroscopy and Hebb–Wagner polarization method. Further, ionic transference numbers

were estimated from the electronic and ionic conductivities: $t_{Li^+} = \frac{\sigma_{Li^+}}{\sigma_e + \sigma_{Li^+}}$ [ACS Energy

Lett. **3**, 1056 (2018), *Phys. Chem. Chem. Phys.* **13**, 19378 (2011)]. As shown in Supplementary Table 1, the tape and pellet electrolytes have almost the same ionic and electronic conductivities, and both are single-ion conductors with nearly equal t_{Li^+} values of ~ 1 .

Supplementary Table 1. Comparison of conductivities of LLZTO pellet and tape from the symmetric cells (Au | LLZTO | Au).

Sample	Ionic conductivity [S/cm]	Electronic conductivity [S/cm]	Ionic transference number
LLZTO Pellet	5.1×10^{-4}	1.46×10^{-9}	0.999997
LLZTO Tape	4.4×10^{-4}	2.8×10^{-9}	0.999993

In addition, we measured the cyclic voltammograms from the Au | LLZTO | Li cells to compare the electrochemical stability windows of the pellet and tape electrolytes, as shown in Supplementary Figure 1. Both electrolytes were stable up to 6 V vs. Li/Li⁺ without exhibiting any decomposition reaction peak, which is consistent with that reported in the literature on electrochemical stability of LLZO electrolytes [*Adv. Energy. Mater.* **6**, 1501590 (2016), *ACS Energy. Lett.* **2**, 462 (2017)].

We also examined the XRD patterns of pellet and tape electrolytes to identify the crystal structure. The XRD patterns were collected using a PANalytical (Empyrean) diffractometer with Cu K α radiation ($\lambda = 1.5406 \text{ \AA}$). Both diffraction patterns, presented in Supplementary Figure 2, reveal cubic garnet structures with trace impurities such as Li₂ZrO₃.

These results evidence that the pellet and tape electrolytes do not significantly differ in terms of their electrochemical properties and crystal structures. In response to your valuable comment,

we have added further details on the basic properties of pellet and tape in **Supplementary information** and added an explanation in the manuscript, as follows:

Supplementary Figure 1. Cyclic voltammograms of pellet and tape measured from Au | LLZTO | Li cells at a scan rate of 0.2 mV s^{-1} within the voltage range of 2–6 V vs. Li/Li^+ .

Supplementary Figure 2. X-ray powder diffraction (XRD) patterns of pellet and tape electrolytes.

A thin electrolyte film was prepared via tape casting. LLZTO powder was added to a mixed solvent of toluene and isopropanol, and was then mixed for approximately 1 h. Fish oil, polyvinyl butyral, and butyl benzyl phthalate were added to the slurry as a binder and dispersant. The resulting slurry was removed as a thin film on a Mylar sheet, and then dried at 120 °C for 1 h. The green tape was sintered at 1100 °C for 2 h.

Revised text

A thin electrolyte film was prepared via tape casting. LLZTO powder was added to a mixed solvent of toluene and isopropanol and was then mixed for approximately 1 h. Fish oil, polyvinyl butyral, and butyl benzyl phthalate were added to the slurry as a binder and dispersant. The resulting slurry was removed as a thin film on a Mylar sheet, and then dried at 120 °C for 1 h. The green tape was sintered at 1100 °C for 2 h.

The ionic and electronic conductivities of the 350- μm -thick pellet and 74- μm -thick tape solid electrolytes were measured from the symmetric cells (Au | LLZTO | Au) by alternating current (AC)-impedance spectroscopy and Hebb–Wagner polarization method. As shown in Supplementary Table 1 and Supplementary Figure 1, the pellet and tape electrolytes have almost the same ionic and electronic conductivities, electrochemical stability windows, and ionic transference numbers ($t_{\text{Li}^+} = \sim 1$). Furthermore, the XRD patterns in Supplementary Figure 2 reveal that both pellet and tape have cubic garnet structures. Therefore, there is no significant difference between the material properties of the 350- μm -thick pellet and 74- μm -thick tape solid electrolytes.

2) *It is confusing for symbol. Ag-coated and Ag-C or AgC in the MS. It needs to distinguish clearly.*

Author reply:

In the manuscript, “Ag-coated LLZTO” refers to Ag-coated solid electrolyte (described in page 11), whereas “Ag–C” or “AgC” represents the composite of silver and carbon, which is used as an interlayer. For clarity and unambiguity, we have used the symbol “AgC” to indicate silver and carbon composite throughout the manuscript.

3) The cathode material is NCM333, it is better describing it more clearly.

Author reply:

In accordance with your suggestion, we have modified the symbol of cathode material “NCM” to be “NCM333” throughout the manuscript.

4) In Page 5, IL amount is being infiltrated in cathode. What is the exact amount IL being uptake by cathode?

Author reply:

To determine the exact amount of infiltrated ionic liquid (IL) in the cathode, we removed the residual catholyte solution on the cathode surface using Kimwipes and then measured the weight change in the IL-infiltrated cathode. The amount of IL uptake by the cathode was estimated to be ~7 wt% relative to the cathode. We have specified this in the *Methods* section on page 5 of the revised manuscript as follows.

Figure R1. Optical images of the ionic-liquid-infiltrated cathode before and after removing the residual solutions on its surface.

Page 5

Original text

The catholyte solution with 20wt% amount relative to the cathode was infiltrated into the cathode in a vacuum for 2 h. We put the ionic liquid-infiltrated cathode into the cathode side of the LLZTO in a pouch cell, and then sealed the cell under vacuum (750 Torr).

Revised text

The catholyte solution (20 wt% relative to the cathode) was infiltrated into the cathode in a vacuum for 2 h. **When the residual solution on the cathode surface was removed using Kimwipes, the solution uptake by the cathode was 7 wt%.** We placed the ionic-liquid-infiltrated cathode on the cathode side of the LLZTO in a pouch cell and then sealed the cell under vacuum (750 Torr).

5) In Fig 1, the model displays look like anode free system without Li metal. But, the authors still applied 20 um Li metal in Full cell. It needs to explain what is the difference?

Author reply:

As pointed out by the reviewer, we identified the preferable Li deposition site in the anode-free model (Figure 1) using DFT calculation and further confirmed current collector side plating in the anode-free cell system (Figure 2). The reason behind the application of 20 μm Li metal in full cell tests is poor capacity retention without Li anode (Supplementary Figure 5). The anode-free cell exhibited a low Coulombic efficiency of 56% and a low discharge capacity of less than 2.0 mAh/cm^2 even after the first cycle of 0.3 mA/cm^2 operation, whereas the use of Li metal led to a high discharge capacity of ~ 2.5 mAh/cm^2 at a high current density of 1.6 mA/cm^2 after 800 cycles. Any pre-existing ductile Li could have diffused into the AgC interlayer under 250 MPa cold isostatic pressing during cell manufacturing, thereby filling the inherent voids or pores in the interlayer. This could potentially generate a percolating Li transport pathway in the AgC interlayer and help to maintain the transport pathway and interface contact without the aid of stack pressure during cell operation.

We have added this description to the manuscript and galvanostatic voltage profiles as Supplementary Figure 5.

Supplementary Figure 5. Galvanostatic voltage profiles of the hybrid electrolyte NCM333-Li full cells with Li-free anode interlayer (Ag/AgC).

Page 12

Original text

For the hybrid cell, Li metal was initially placed at the anode to promote Li nucleation, and an ionic liquid was used at the cathode for enhanced interface wetting and utilization.

Revised text

For the hybrid cell, an ionic liquid was used at the cathode for enhanced interface wetting and utilization. In addition, Li metal was initially placed at the anode to improve capacity retention. As described in Supplementary Figure 5, the anode-free cell exhibited a low Coulombic efficiency of 56% and a low discharge capacity of less than 2.0 mAh/cm² even after the first cycle of 0.3 mA/cm² operation. Any pre-existing ductile Li could have diffused into the AgC interlayer under 250 MPa CIP during cell manufacturing, thereby filling the inherent voids or pores in the interlayer. This could potentially generate a percolating Li transport pathway in the AgC interlayer and help to maintain the transport pathway and interface contact without the aid of stack pressure during cell operation.

6) *Symmetric cell with IL and without IL exp. needs to explain or prove the tolerance of Li dendritic generation.*

Author reply:

In accordance with the reviewer's comment, we measured the critical current densities (CCDs) of the symmetric cells without ionic liquid to demonstrate the improved tolerance to Li dendrite formation in the presence of the Ag/AgC interlayer. Supplementary Figure 6 depicts the results of galvanostatic cycle tests on the symmetric Li | LLZTO | Li and Li | AgC | Ag | LLZTO | Ag

| AgC | Li cells at 25 °C with increasing current densities from 0.2 to 1.6 mA/cm² with a step size of 0.2 mA/cm². The symmetric cells were cycled twice with 30 min of lithium plating/stripping at each current density step. The curves imply that the cell employing the Ag/AgC interlayer could be cycled at a high current density of 1.6 mA/cm², whereas that without the interlayer short-circuited at 0.4 mA/cm². The symmetric cell results prove that the Ag/AgC interlayer is instrumental in preventing Li dendrite formation at the anode/solid electrolyte interfaces, which is consistent with the full cell results shown in Figure 5.

The relevant results have been added to the Supplementary Information, and the explanation has been added to the manuscript.

Supplementary Figure 6. Galvanostatic cycling of the symmetric cells with increasing current densities from 0.2 to 1.6 mA/cm² with a step size of 0.2 mA/cm².

Original text

Accordingly, the charging/discharging curve (Fig. 5c, d) shows that the hybrid cell without Ag failed at a current density of 1 mA/cm² by short-circuiting, while the cell with the Ag

layer was capable of operating at a high current density of 1.6 mA/cm² without short-circuiting.

Revised text

Accordingly, the charging/discharging curve (Fig. 5c, d) shows that the hybrid cell without Ag failed at a current density of 1 mA/cm² by short-circuiting, while the cell with the Ag layer was capable of operating at a high current density of 1.6 mA/cm² without short-circuiting. Galvanostatic cycling tests performed on the symmetric Li | AgC | Ag | LLZTO | Ag | AgC | Li cell without ionic liquid confirmed the short-circuit tolerance of the Ag/AgC interlayer. The symmetric cell with the Ag/AgC interlayer cycled at 1.6 mA/cm² without short-circuiting, whereas the cell without the interlayer short-circuited at 0.4 mA/cm², as shown in Supplementary Figure 6.

7) The authors needs to explain why Ag in coating layer and Ag-C interlayer are moving (it is Li-Ag alloy is being formed during the charge). How come? It need to compare the previous paper results, the Ag is moving to Cu current collector.

Ref: “High-energy long-cycling all-solid-state lithium metal batteries enabled by silver–carbon composite anodes”, *Nat Energy*, <https://doi.org/10.1038/s41560-020-0575-z>

Author reply:

We agree with the reviewer’s observation; Ag moved toward the lithium metal to form Li–Ag alloy, as shown in the EDS map in Figure 4c. A similar phenomenon has been reported for a Li-free AgC interlayer in a previous study [*Nat. Energy* **5**, 299 (2020)], wherein Li was plated at the current collector interface during charging. In both cases, Ag apparently entered the

plated lithium during charging and formed a Li–Ag alloy. According to S.Y. Kim and J. Li’s paper [*Energy Mater. Adv.* **2021**, 1519569 (2021)], the driving force for Ag transport toward Li can be explained in terms of thermodynamic chemical energy in the Li–Ag alloy. Based on the chemical potential changes in the Li–Ag alloy system, the entry of Ag atoms into the Li–Ag alloy is expected to be more thermodynamically stable than that of Li atoms. Therefore, the Ag atoms in the coating layer and AgC interlayer tend to migrate toward the plated metallic Li during charging, forming Li–Ag alloy phases. Nevertheless, we would like to emphasize that a part of Ag in the coating layer remains at the interface between LLZTO and AgC interlayer after 100 cycles (Figure 4c), contributing to the prolonged interfacial contact.

This explanation has been added in the revised manuscript, and the related work has been cited.

Page 12

Original text

The SEM image and EDS data (Fig. 4c) show that Ag remained in the layer form even after 100th cycle at the carbon-LLZTO interface, although some Ag particles were observed to be distributed in the Li metal owing to their thermodynamic preference for the formation of Li–Ag alloy during cycling^{23,30}.

Revised text

The SEM image and EDS data (Figure 4c) show that Ag remained in the layer form even after 100th cycle at the carbon-LLZTO interface, although some Ag particles were observed to be distributed in the Li metal owing to their thermodynamic preference for the formation of Li–Ag alloy during cycling^{23,31}. Diffusion of Ag into Li has been reported in literature^{23,32}.

Kim and Li attributed this phenomenon to the chemical potential change due to Li–Ag alloying³². According to their estimation, the chemical potential change in Ag is expected to be much higher than that in Li. Therefore, the Ag atoms in the coating layer and AgC interlayer migrate toward the plated metallic Li during charging, forming Li–Ag alloy phases.

32. Kim, S. Y. & Li, J. Porous Mixed Ionic Electronic Conductor Interlayers for Solid-State Batteries. *Energy Mater. Adv.* **2021**, (2021).

8) In Fig. 4, The I-V curves are needed to support the long-term cycling performance?

Author reply:

In accordance with the reviewer’s suggestion, we have added I–V curves pertaining to long-term cycling performance (after the 400th, 700th, and 800th cycles) to Figure 5d.

Figure 5d

Original manuscript

Revised manuscript

9) In Fig. 5, the EIS spectra are need a suitable EIS Circuit Model, and please compare R_b , R_{ct} , W parameters in detail.

Author reply:

Following the suggestions of the reviewer and the editor, we calculated the ohmic resistance (R_b), interfacial resistance (R_{ct}), and Warburg resistance (W_R) by complex nonlinear least-squares fitting of the experimental impedance spectra in Figure 5b based on a typical equivalent circuit of the NCM-Li full cell [*J. Power Sources* **160**, 1342 (2006), *J. Electrochem. Soc.* **159**, A1324 (2004), *J. Mater. Chem.* **21**, 10777 (2011)]. We have added images of the equivalent circuit and fitting curves in Figure 5b and listed the resultant fitting values to Supplementary Table 4 to facilitate a better comparison of R_b , R_{ct} , and W_R .

Supplementary Table 4. Fitting parameters determined from CNLS fitting of the measured impedance spectra in Figure 5b.

Fitting parameters	Anode interlayer	
	AgC	Ag/AgC
R_b [Ω cm ²]	11.5	10.7
R_l [Ω cm ²]	39.4	22.8
R_2 [Ω cm ²]	27.0	9.7
$R_{ct} = R_l + R_2$ [Ω cm ²]	66.4	32.5
CPE ₁ -T	1.17×10^{-6}	1.29×10^{-6}
CPE ₁ -n	0.96	0.97
W_R [Ω cm ²]	58.0	44.4
W-T	0.21	0.17
W-n	0.41	0.42
CPE ₂ -T	2.03×10^{-6}	5.41×10^{-6}
CPE ₂ -n	0.72	0.70
chi-square	1.58×10^{-5}	2.32×10^{-5}

Page 14

Original text

Because the predominant issue for utilizing the LLZTO tape was the contact between LLZTO and the AgC interlayer (Fig. 3a, 3b), we measured the AC impedance spectra to

investigate the change in interfacial resistance. The Nyquist plot of the impedance spectra (Fig. 5b) shows that the Ohmic losses are virtually identical for the both cells, whereas the high-frequency arc associated with the charge-transfer reactions at the cathode–anode interfaces decreases significantly when Ag is present on the LLZTO surface. Assuming that the cathode interfacial kinetics is unchanged in the presence of Ag, the decrease in the high-frequency arc is attributable to the improved charge-transfer kinetics at the anode-LLZTO interface due to the increased effective area and the enhanced bonding of anode to LLZTO.

Original Figure 5b

Revised text

Because the predominant issue for utilizing the LLZTO tape was the contact between LLZTO and the AgC interlayer (Figures 3a, 3b), we measured the AC impedance spectra to investigate the change in interfacial resistance. Figure 5b shows that the Nyquist plot of the impedance spectra consists of two depressed arcs, which slightly overlap in the high-frequency range, Warburg impedance, and a capacitive line (blocking region) in the low-frequency range^{33–35}. The equivalent circuit, shown in the inset of Figure 5b, was implemented to quantitatively analyze the experimental impedance spectra. The Ohmic

resistance (R_b), overall interfacial resistance ($R_{ct} = R_1 + R_2$), and Warburg resistance (W_R) were evaluated using the complex nonlinear least-squares (CNLS) fitting method; the values are listed in Supplementary Table 4. There are no drastic changes in the R_b and W_R for both cells; the R_{ct} , which is associated with the charge-transfer reactions at the cathode–anode interface, significantly decreases from 66.4 to 32.5 $\Omega \text{ cm}^2$ when Ag is present on the LLZTO surface. Assuming that the cathode interfacial kinetics is unchanged in the presence of Ag, the decrease in the R_{ct} is attributable to the improved charge-transfer kinetics at the anode–LLZTO interface due to the increased effective area and the enhanced bonding of anode to LLZTO.

33. Li, D., Kato, Y., Kobayakawa, K., Noguchi, H. & Sato, Y. Preparation and electrochemical characteristics of $\text{LiNi}_{1/3}\text{Mn}_{1/3}\text{Co}_{1/3}\text{O}_2$ coated with metal oxides coating. *J. Power Sources* **160**, 1342–1348 (2006).

34. Shaju, K. M., Rao, G. V. S. & Chowdari, B. V. R. Influence of Li-Ion Kinetics in the Cathodic Performance of Layered $\text{Li}(\text{Ni}_{1/3}\text{Co}_{1/3}\text{Mn}_{1/3})\text{O}_2$. *J. Electrochem. Soc.* **151**, A1324 (2004).

35. Huang, Z.-D. *et al.* Microscopically porous, interconnected single crystal $\text{LiNi}_{1/3}\text{Co}_{1/3}\text{Mn}_{1/3}\text{O}_2$ cathode material for Lithium ion batteries. *J. Mater. Chem.* **21**, 10777–10784 (2011).

Revised Figure 5b

10) It seems that the I-V curve results are not consistency with the cycling test results. The constant current section is quite short after 100 cycles. Why? Fig. 6(b) is using capacity of mAh/cm², but in Fig. 6(c), the capacity is using mAh?

Author reply:

As the reviewer has mentioned, the CC region diminished with increasing cycling in Figure 5d owing to the increase in cell polarization at a 1.6 mA/cm² operation. The polarization increase at 0.5 C-rate might be associated with the degradation of the anode/electrolyte interface. The contact between the anode and electrolyte could be moderately impaired during electrochemical lithium plating and stripping because of dynamic volume expansion and contraction at the anode. In addition, as we discussed in the reply to comment #5, some coated Ag might migrate to Li metal during charging, reducing the bonding strength between the anode and electrolyte. This partial interfacial degradation could result in the loss of active sites for the charge-transfer reaction with increasing cycles, thereby increasing cell polarization.

Nevertheless, we would like to emphasize that our interlayer is instrumental in regulating Li deposition site after hundreds of cycles, effectively preventing short circuit.

In Figure 6c, we had expressed in terms of the overall cell capacity (mAh) rather than areal capacity (mAh/cm²) to emphasize the relatively larger capacity therein than that of the coin-sized cell. To avoid any confusion and ensure consistency, this modification has been applied to Figure 6b as well.

The following modifications have been performed in the manuscript:

Page 14

Original text

Owing to the adhesion of Ag, our interlayer could successfully protect LLZTO from dendrite penetration, which helped realize a high volumetric energy density and an excellent long-term cycle stability for 800 cycles (Fig. 5e). Considering that our cell was operated at room temperature without external pressure, these results pave the way for the use of LMBs in various fields, such as in mobile and wearable devices, that require low-temperature operation and limited volume. Additionally, this type of cell can potentially deliver a considerable energy density of 680 Wh/L assuming our hybrid battery configuration as one-unit cell in pack, which is promising for practical applications.

Revised text

Moreover, the overpotential gradually increased upon cycling, diminishing the CC charge region in Figure 5d. This might be associated with the physical degradation of the anode/electrolyte interface, originating from the local stress due to volume change upon cycling or Ag diffusion into Li metal. Nevertheless, our interlayer could successfully protect

LLZTO from dendrite penetration, which helped realize a high volumetric energy density and an excellent long-term cycle stability for 800 cycles (Fig. 5e). Considering that our cell was operated at room temperature without external pressure, these results pave the way for the use of LMBs in various fields, such as in mobile and wearable devices, that require low-temperature operation and limited volume. Additionally, this type of cell can potentially deliver a considerable energy density of 680 Wh/L assuming our hybrid battery configuration as one-unit cell in pack (Supplementary Table 5), which is promising for practical applications.

Original Figure 6b

Revised Figure 6b

11) There are a lot of syntax errors; like 12-mA-capacity cell in Page 14, line380, and 3.4 mA/cm in page 3, line 64, etc.

Author reply:

We have thoroughly rechecked the manuscript for such syntax errors and rectified them accordingly.

Revised text

As shown in Fig. 6a, the larger area cathode was used in this cell and the 12-mAh-capacity cell was tested at room temperature.

Using a AgC composite interlayer and sulfide solid electrolyte, a high reversibility for ~1000 cycles was demonstrated in a pouch cell with a high current density of 3.4 mA/cm²²³.

12) Please describe it more detail, How you get ED of 680 Wh/L?

Author reply:

We calculated the volumetric energy density based on a single cell, including the package films. In particular, we used a reduced electrolyte thickness (~74 μm) after the acid treatment instead of the initial thickness (80 μm) for the calculation. The details on energy density calculations have been summarized in Supplementary Table 5.

Supplementary Table 5. Parameters considered for estimating energy densities of the full cells.

Cathode	Composition	NCM333
	Areal capacity (mAh/cm^2)	3.2
	Thickness (μm)	50
Solid electrolyte	Initial thickness (μm)	80
	Thickness after acid treatment	74
Anode	Ag/AgC interlayer (μm)	6
	Li thickness (μm)	20
Current collector	Al for cathode (μm)	9
	Cu for anode (μm)	5
Total thickness (μm)		164
Cell area (cm^2)		30.2
Cell volume including package film (L)		0.540
Discharge V_{ave} (V)		3.8
Energy density (Wh/L)		680

Original text

Additionally, this type of cell can potentially deliver a considerable energy density of 680 Wh/L assuming our hybrid battery configuration as one-unit cell in pack, which is promising for practical applications.

Revised text

Additionally, this type of cell can potentially deliver a considerable energy density of 680 Wh/L assuming our hybrid battery configuration as one-unit cell in pack (Supplementary Table 5), which is promising for practical applications.

Reviewer #2

Comments:

In this work, the authors investigated a novel lithium metal -garnet type solid electrolytes (Ta-doped LLZO) Interface where Ag-C composite interlayer inserted in between. The working principle and formation mechanism of the interface/battery have been specifically investigated. Although the experimental data are systematic and the evidence is sufficient, however, I found the presentation of the manuscript quite simple, even a bit poor, the computational modeling details are unclear. The central topic of the article seems to be unclear enough caused by the unreasonable arrangement of the article, which is unfavorable for readers to understand the work. Some arguments are not logical and compelling, and the novelty of the article could not match the high standard of this high-impact journal. Therefore, the article is suggested to be reorganized, and major revisions are required. The following comments might be helpful.

Author reply:

We appreciate the reviewer for their elaborate review of our manuscript and helpful comments.

Their comments, followed by our point-by-point responses, are shown below.

1) Although the author has described the definition of adhesion energy, a table with the energy values for different simulations will be helpful.

Author reply:

We agree with your valuable suggestion. Accordingly, we have inserted a table listing the calculated adhesion energies in the Supplementary Information. The adhesion energies shown in the Supplementary Table 2 were used to determine the favorable Li deposition site in “Role of interlayer to regulated Li deposition site” section (Figure 1, lines 218–225).

Interfaces	LLZTO/C	LLZTO/Li	Cu/C	Cu/Li	Fe/C	Fe/Li	LPSCI/C	LPSCI/Li
Adhesion energy (J/m ²)	-0.330	-0.708	-0.595	-1.485	-0.698	-2.051	-0.167	-0.854

Supplementary Table 2. Calculated adhesion energies of various interfaces.

Page 8

Original text

To assess the interface energies, atomic models were built for each interface, as shown in Fig. 1c, for the LLZTO/Li interface, and the work of adhesion (W_{ad}) was calculated accordingly (refer to Computational Details section for interface modeling process).

Revised text

To assess the interface energies, atomic models were built for each interface, as shown in Figure 1c, for the LLZTO/Li interface, and the work of adhesion (W_{ad}) was calculated accordingly (refer to Computational Details section for interface modeling process).

Supplementary Table 2 summarizes the calculated adhesion energies of various interfaces.

2) On page 6, lines 163-164; author mentioned shifting one of the surfaces to determine stable alignment between two surfaces after relaxation. It is not clear on what basis the stable alignment was decided.

Author reply:

As the reviewer has mentioned, the process of optimizing interface structures needs further clarification. Interface relaxation is more complicated than bulk relaxation; the optimal distance between two surfaces should be evaluated. Furthermore, the most stable lateral alignment between two surface structures should also be identified. Therefore, it is difficult to determine the ground state using simple one-step relaxation. Hence, we conducted a two-step calculation.

First, we calculated the total energy of the interface models with different distances between two surface slabs. In this step, we could determine the optimal interdistance, which results in the lowest total energy. Next, we explored various lateral alignments by shifting a surface slab. Here, the magnitude of two-dimensional lateral shift was determined by considering the

symmetry of the atomic arrangement. Similar to the first step, we could obtain the optimal lateral alignment by selecting the lowest-energy structure.

To clarify the process of optimizing interface structures, we have considered the LLZTO/Li interface (Figure 1c) as an example and have summarized the relaxation process in Figure R2.

Figure R2. Schematics showing the optimization process of the LLZTO/Li interface

3) *The authors claimed/showed long cycle stability of the interface (800 cycles at room temperature) however there is no computational support for this point. To investigate the interactions between two surfaces, for example between Li and LLZTO, the ab initio molecular dynamics (AIMD) simulations at operating temperature are needed. Did the author think in this direction?*

Author reply:

As demonstrated in the manuscript, the carbon-based interlayer (AgC) could effectively prevent Li dendrite growth by regulating the Li deposition site (Figure 2). A major challenge was the weak adhesion between the AgC interlayer and LLZTO tape (Figures 3a, 3c), resulting in anode instability and poor cycle stability (Figure 5c). To address this issue, we sputtered an Ag layer between LLZTO and AgC to enhance the interfacial adhesion, and the effect of Ag was demonstrated both experimentally and theoretically (Figures 3b, 3d, 3e). Our analyses implied that the long cycle stability of the interface is attributed to the enhanced adhesion due to the sputtered Ag layer, validated by experiments and calculations.

We agree with the reviewer that room-temperature AIMD simulations are helpful for directly understanding the interfacial stability. However, the timescale of room-temperature AIMD may not be long enough for a feasible evaluation of the interfacial stability. To prove the long-term interfacial stability, room-temperature AIMD simulation must be run over a long timescale (~nanoseconds). However, the computational cost for long-term AIMD is extremely high, considering the complexity of the trilayer interface model of LLZTO/Ag/AgC. Even for bulk Li ion conductor structures, which are much simpler than the LLZTO/Ag/AgC model, AIMD simulations are conducted at high temperatures (~1000 K) because it is difficult to detect substantial Li jumps at room temperature. Therefore, to the best of our knowledge, it would be difficult to evaluate the long-term interfacial stability by room-temperature AIMD simulations.

Instead, we have computationally demonstrated the origin of long cycle stability via interfacial adhesion energy calculations, whereby the enhanced adhesion due to the Ag coating layer enables long-term interface integrity.

4) Another point is LLZO is partial disordered material, how did the author build that structure and how Ti atoms doped in Zr sites.

Author reply:

As the reviewer pointed out, two Li sites in LLZTO are not fully occupied [*Chem. Lett.* **40**, 60 (2011)]. In addition to the Li/vacancy disorder at a Li site, Ta substitution at a Zr site induces Zr/Ta site disorder. With a view to addressing this issue, we first generated various disordered structures and performed Ewald summation using *pymatgen* package to screen 30 lowest-energy structures. Here, the Li occupancies at both Li sites (24d, 96h) were set to the values reported by Awaka *et al.* [*Chem. Lett.* **40**, 60 (2011)]. Then, we conducted DFT calculations for the screened structures to determine the lowest-energy structure.

We have added a description in the Methods section, as follows:

Page 6

Original text

Interface models with sufficiently thick ($> 15 \text{ \AA}$) vacuum slabs were constructed to describe various interfaces in LMBs.

Revised text

To model the atomic structure of partially disordered LLZTO, we first generated various disordered structures and performed Ewald summation using *pymatgen* package²⁸ to screen 30 lowest-energy structures. Here, the Li occupancies at both Li sites (24d, 96h) were set to the values reported by Awaka *et al.*²⁹. Then, we conducted DFT calculations for the screened structures to determine the lowest-energy structure.

Interface models with sufficiently thick ($> 15 \text{ \AA}$) vacuum slabs were constructed to describe various interfaces in LMBs.

29. Awaka, J. *et al.* Crystal Structure of Fast Lithium-ion-conducting Cubic $\text{Li}_7\text{La}_3\text{Zr}_2\text{O}_{12}$. *Chem. Lett.* **40**, 60–62 (2010).

5) *No figures related to the simulations of the LLZTO-Ag-Graphite.*

Author reply:

As the reviewer has pointed out, atomistic model structures for LLZTO/Ag/Graphite (Figure 3d) were not included in the manuscript. To clarify the interfacial adhesion energy calculations for the LLZTO/Ag/graphite structure, we have added the atomic model structures of the LLZTO/Ag and Ag/graphite interfaces.

Supplementary Figure 3. Atomic models of (a) Ag/C interface and (b) LLZTO/Ag interface.

Page 10

Original text

Calculated work of adhesions between LLZTO and Ag, and that between Ag and graphite are -0.80 and -0.56 J/m², respectively (Fig. 3d)

Revised text

Calculated work of adhesions between LLZTO and Ag, and that between Ag and graphite are -0.80 and -0.56 J/m², respectively (Figure 3d and Supplementary Figure 3).

6) Page 8, lines 221-223 are unclear. Have the simulations with graphite and Cu or Fe are performed? There is no figure shown detailing it. Moreover, the meaning of the numbers in J/m² are not clearly mentioned.

Author reply:

We agree that lines 221-223 are unclear in the original manuscript owing to the lack of detailed information. As mentioned in the manuscript, we performed DFT calculations to evaluate the adhesion energies of various interfaces and predict the favorable Li deposition site. Figure 1c shows LLZTO/Li interface models as an example, and several other interface models, including graphite with Cu or Fe, were built to calculate the adhesion energies. For example, the Fe/graphite interface model is shown in Figure R3.

On the basis of the adhesion energies of various interfaces (Supplementary Table 2 and our reply to reviewer's comment #1), we predicted the favorable site for Li deposition. As described in Figures 1a and 1b, two Li deposition sites exist, and accordingly, different interfaces are formed based on the Li plating site. For example, if Li is plated between LLZTO and the interlayer (Figure 1a), LLZTO/Li, Li/C, and C/Cu function as the interfaces. On the other hand, if Li is plated between the interlayer and current collector (Figure 1b), the interfaces in the cell are LLZTO/C, Li/C, and Li/Cu. Therefore, by comparing the adhesion energies of those different interfaces, we could predict the favorable Li deposition site. Supplementary Table 3 summarizes our results for various electrolyte/current collector combinations. Here, the values in J/m^2 (lines 223-225) indicate the extent of preference for Li to be plated between the electrolyte and current collector. For instance, if a graphite interlayer is used with a LLZTO electrolyte and Cu current collector, Li will favorably be plated at the Cu/graphite interface with an energy of $(-1.303 \text{ J/m}^2) - (-1.815 \text{ J/m}^2) = 0.512 \text{ J/m}^2$.

We have revised a figure and added following explanation and tables to the manuscript for clarification.

Interfaces	LLZTO/C	LLZTO/Li	Cu/C	Cu/Li	Fe/C	Fe/Li	LPSC/C	LPSC/Li
Adhesion energy (J/m ²)	-0.330	-0.708	-0.595	-1.485	-0.698	-2.051	-0.167	-0.854

Supplementary Table 2. Calculated adhesion energies of various interfaces.

Cases		Li plating at electrolyte/interlayer interface (Figure 1a) Interfaces: Collector/Interlayer + Li/Electrolyte	Li plating at collector/interlayer interface (Figure 1b) Interfaces: Collector/Li + Interlayer/Electrolyte
Sum of interfacial adhesion energies (J/m ²)	Electrolyte: LLZTO Current collector: Cu	$(-0.595) + (-0.708) = (-1.303)$	$(-1.485) + (-0.330) = (-1.815)$
	Electrolyte: LLZTO Current collector: Fe (SUS)	$(-0.698) + (-0.708) = (-1.406)$	$(-2.051) + (-0.330) = (-2.381)$
	Electrolyte: Li ₆ PS ₅ Cl Current collector: Cu	$(-0.595) + (-0.854) = (-1.449)$	$(-1.485) + (-0.167) = (-1.652)$

Supplementary Table 3. Sums of interfacial adhesion energies in two different Li plating scenarios. Adhesion energy of the Li/interlayer interface was omitted because it is accounted for in both scenarios. Detailed schematics of the two scenarios are shown in Figure 1.

Figure R3. Atomic model of the Fe/graphite interface

Figure 1a, 1b

Original figure

Revised figure

Page 8

Original text

As shown in Fig. 1a and 1b, the internal arrangement of cell components after charging varies with the Li plating site, leading to a distinct interface for each configuration. Therefore, by comparing total interface energies, we could predict the thermodynamically favorable Li deposition site. To assess the interface energies, atomic models were built for each interface, as shown in Fig. 1c, for the LLZTO/Li interface, and the work of adhesion (W_{ad}) was calculated accordingly (refer to Computational Details section for interface modeling process). Using graphite as an interlayer and Cu or Fe (representing SUS) as a current collector, our DFT calculations suggest that the latter case (Fig. 1b) is the more favorable state by $\sim 0.51 \text{ J/m}^2$ and $\sim 0.98 \text{ J/m}^2$, respectively.

Revised text

As shown in Figures 1a and 1b, the internal arrangement of cell components after charging varies with the Li plating site, leading to a distinct interface for each configuration. For example, when LLZTO is used as the solid electrolyte and Cu as the current collector along with graphite interlayer, two Li deposition sites exist: LLZTO/graphite interface and graphite/Cu interface. If Li is plated between LLZTO and the interlayer (Figure 1a), LLZTO/Li, Li/C, and C/Cu function as the interfaces. On the other hand, if Li is plated between the interlayer and current collector (Figure 1b), the interfaces in the cell are LLZTO/C, Li/C, and Li/Cu. Therefore, by comparing total interface energies, we could predict the thermodynamically favorable Li deposition site. To assess the interface energies, atomic models were built for each interface, as shown in Figure 1c, for the LLZTO/Li interface, and the work of adhesion (W_{ad}) was calculated accordingly (refer to Computational Details section for interface modeling process). Supplementary Table 2 summarizes the calculated adhesion energies of various interfaces. From these adhesion energies, we could compare the sums of the interfacial adhesion energies between the scenarios and predict the preferable Li deposition site (Supplementary Table 3). For example, in the scenario of graphite interlayer and Cu or Fe (representing SUS) current collector, our DFT calculations suggest that the latter case (Figure 1b) is the more favorable state by ~ 0.51 or ~ 0.98 J/m², respectively.

7) lines 226-254 the authors discussed and concluded about Li transport through the interlayer without any computational/theoretical evidence. Li transport/migration barriers through the

Li/LLZO have been reported in the paper (journal of The Electrochemical Society, (2020) 167 100537). The question coming is what is the Li barrier in this scenario?

Author reply:

In lines 226-254, we have discussed the importance of fast Li transport in inducing Li plating between the current collector and interlayer. As described in the Discussion section, the reduced Li at the solid electrolyte/interlayer interface should be quickly transported through the interlayer to prevent Li accumulation at the interface during charging. In this regard, understanding the Li transport mechanism through the interlayer is critical for determining methods to enhance the Li transport kinetics. Referred paper [*J. Electrochem. Soc.* **167**, 100537 (2020)] reported Li ion diffusivity within the bulk and grain boundary domains of LLZO using MD simulations and further discussed the implications based on the bulk/grain boundary inhomogeneity. The Li transport through the interlayer in this study could be understood in similar method, but we expect the Li transport mechanism through the interlayer to differ from that through LLZO. Unlike LLZO, AgC interlayer is a mixed ion–electron conductor. Therefore, during charging, Li ion transported from the solid electrolyte should be reduced at the solid electrolyte/interlayer interface, and the reduced Li metal would then be transported through the interlayer. Hence, we expect that plastic deformation of Li metal, especially creep, is the most probable Li transport mechanism through the interlayer [*Nature* **578**, 251 (2020), *J. Electrochem. Soc.* **166**, A89 (2019)]. It would be challenging to computationally calculate the Li transport barrier via creep; for a simple estimate, LePage *et al.* reported the activation energy for creep of pure Li metal (Q_c) as $37 \pm 6 \text{ kJ mol}^{-1}$, which is $\sim 0.38 \text{ eV}$ [*J. Electrochem. Soc.* **166**, A89 (2019)].

We have added the following description to the manuscript for clarification:

Original text

This indicates that the thermodynamically favorable Li deposition site is between the current collector and interlayer, unless there is no kinetic barrier for Li transport through the interlayer.

Revised text

This indicates that the thermodynamically favorable Li deposition site is between the current collector and interlayer, unless there is a kinetic barrier for Li transport through the interlayer.

During charging, Li^+ transported from the solid electrolyte would be reduced at the interface between the solid electrolyte and mixed ion–electron conducting interlayer. If the time for the reduced Li to move toward the current collector through the interlayer via creep is insufficient, Li would be undesirably plated near the solid electrolyte, although it is thermodynamically unfavorable. In other words, Li transport through the interlayer should be faster than Li influx from the cathode side, which is dependent on the charge rate. Therefore, it is crucial to enhance Li transport kinetics through the interlayer for fast charge/discharge.

8) The adhesion energy and charge calculations did not fully interpret the thermodynamic and the origin of the preferable Li deposition process.

Author reply:

We believe that this comment is in line with comment #6 since both are related to the logic behind predicting favorable Li deposition site using adhesion energy calculations. In response

to comment #6, we have added Supplementary Tables 2 and 3 and attempted clarifying our interpretation of the thermodynamic origin of preferable Li deposition.

When an interlayer is used at the anode, two Li deposition sites exist, namely, solid electrolyte/interlayer interface and interlayer/current collector interface. Different interfaces are generated in each case (Figure 1a, 1b), and our prime reasoning is to predict the thermodynamically favorable Li deposition site by comparing the sums of interfacial adhesion energies between scenarios. For example, when LLZTO is used as the solid electrolyte and Cu as the current collector along with graphite interlayer, two possible Li deposition sites exist: LLZTO/graphite interface and graphite/Cu interface. According to our calculations for the scenarios, the sum of the interfacial adhesion energies is more negative when Li is plated between graphite and Cu current collector. Therefore, we conclude that Li is preferably deposited at the graphite/Cu interface.

We have revised a figure and added following explanation and tables to the manuscript for clarification.

Tables added:

Interfaces	LLZTO/C	LLZTO/Li	Cu/C	Cu/Li	Fe/C	Fe/Li	LPSCI/C	LPSCI/Li
Adhesion energy (J/m ²)	-0.330	-0.708	-0.595	-1.485	-0.698	-2.051	-0.167	-0.854

Supplementary Table 2. Calculated adhesion energies of various interfaces.

Cases		Li plating at electrolyte/interlayer interface (Figure 1a)	Li plating at collector/interlayer interface (Figure 1b)
		Interfaces: Collector/Interlayer + Li/Electrolyte	Interfaces: Collector/Li + Interlayer/Electrolyte
Sum of interfacial adhesion energies (J/m ²)	Electrolyte: LLZTO Current collector: Cu	$(-0.595) + (-0.708) = (-1.303)$	$(-1.485) + (-0.330) = (-1.815)$
	Electrolyte: LLZTO Current collector: Fe (SUS)	$(-0.698) + (-0.708) = (-1.406)$	$(-2.051) + (-0.330) = (-2.381)$
	Electrolyte: Li ₆ PS ₅ Cl Current collector: Cu	$(-0.595) + (-0.854) = (-1.449)$	$(-1.485) + (-0.167) = (-1.652)$

Supplementary Table 3. Sums of interfacial adhesion energies in two different Li plating scenarios. Adhesion energy of the Li/interlayer interface was omitted because it is accounted for in both scenarios. Detailed schematics of the two Li plating scenarios are shown in Figure 1.

Figure 1a, 1b

Original figure

Plating at LLZTO/interlayer interface (undesirable)

Plating at collector/interlayer interface (desirable)

Revised figure

Plating at LLZTO/interlayer interface (undesirable)

Interfaces:
Current collector/Interlayer + Interlayer/Li + Li/LLZTO

Plating at collector/interlayer interface (desirable)

Interfaces:
Current collector/Li + Li/Interlayer + Interlayer/LLZTO

Original text

As shown in Fig. 1a and 1b, the internal arrangement of cell components after charging varies with the Li plating site, leading to a distinct interface for each configuration. Therefore, by comparing total interface energies, we could predict the thermodynamically favorable Li deposition site. To assess the interface energies, atomic models were built for each interface, as shown in Fig. 1c, for the LLZTO/Li interface, and the work of adhesion (W_{ad}) was calculated accordingly (refer to Computational Details section for interface modeling process). Using graphite as an interlayer and Cu or Fe (representing SUS) as a current collector, our DFT calculations suggest that the latter case (Fig. 1b) is the more favorable state by $\sim 0.51 \text{ J/m}^2$ and $\sim 0.98 \text{ J/m}^2$, respectively.

Revised text

As shown in Figures 1a and 1b, the internal arrangement of cell components after charging varies with the Li plating site, leading to a distinct interface for each configuration. For example, when LLZTO is used as the solid electrolyte and Cu as the current collector along with graphite interlayer, two Li deposition sites exist: LLZTO/graphite interface and graphite/Cu interface. If Li is plated between LLZTO and the interlayer (Figure 1a), LLZTO/Li, Li/C, and C/Cu function as the interfaces. On the other hand, if Li is plated between the interlayer and current collector (Figure 1b), the interfaces in the cell are LLZTO/C, Li/C and Li/Cu. Therefore, by comparing total interface energies, we could predict the thermodynamically favorable Li deposition site. To assess the interface energies, atomic models were built for each interface, as shown in Figure 1c, for the LLZTO/Li interface, and the work of adhesion (W_{ad}) was calculated accordingly (refer to Computational Details section for interface modeling process). Supplementary Table 2

summarizes the calculated adhesion energies of various interfaces. From these adhesion energies, we could compare the sums of the interfacial adhesion energies between the scenarios and predict the preferable Li deposition site (Supplementary Table 3). For example, in the scenario of graphite interlayer and Cu or Fe (representing SUS) current collector, our DFT calculations suggest that the latter case (Figure 1b) is the more favorable state by ~ 0.51 or ~ 0.98 J/m², respectively.

REVIEWER COMMENTS

Reviewer #1 (Remarks to the Author):

The authors are response the all questions already.

I agree to accept the revised MS to be published in Nat Comm J.

Reviewer #2 (Remarks to the Author):

The authors emphasized and addressed most of my questions however there are three questions below they did not satisfy me.

2) On page 6, lines 163-164;

Could you please clarify what is final lattice mismatch of your LLZTO/Li interface ?

3) The authors claimed/showed long cycle stability of the interface (800 cycles at room temperature) however there is no computational support for this point. To investigate the interactions between two surfaces, for example between Li and LLZTO, the ab initio molecular dynamics (AIMD) simulations at operating temperature are needed. Did the author think in this direction?

I do not agree with the answer. In our computational community, for Li metal anode system, at very least, we all usually can perform AIMD for the interface at $T=300$ K about 20 to 40 ps and hence, I would suggest the authors should perform this step to have a solid conclusions for their paper.

7) lines 226-254 the authors discussed and concluded about Li transport through the interlayer without any computational/theoretical evidence.

The authors can use cNEB method to calculate Li migration barriers at the interface to support the observation.

Reviewer #1

Comments:

The authors are response the all questions already.

I agree to accept the revised MS to be published in Nat Comm J.

Author reply:

We appreciate the reviewer for positive comments on our revised manuscript. Comments of the reviewer were highly helpful to enhance the quality of our work.

Reviewer #2

Comments:

The authors emphasized and addressed most of my questions however there are three questions below they did not satisfy me.

Author reply:

We are grateful to the reviewer for an elaborate review on our manuscript and delivering helpful comments. We agree that the additional points raised by the reviewer were crucial to enhance the quality of our work. Once again, we reproduce the comments of the reviewer, followed by our point-to-point responses to the comments and suggestions of the reviewer.

2) *On page 6, lines 163-164;*

Could you please clarify what is final lattice mismatch of your LLZTO/Li interface ?

Author reply:

We thank the reviewer for pointing out the degree of lattice mismatch of LLZTO/Li interface. Indeed, a lattice mismatch arises when we build an interface, due to the different crystallographic properties of two materials. In our LLZTO (001) / Li (100) interface, we used 4 X 4 Li (100) supercell ($a = 3.427 \text{ \AA}$) to match the (001) surface of LLZTO unit cell ($a = 12.968 \text{ \AA}$). Accordingly, average lattice strain $\bar{\epsilon}$ [*J. Phys.: Condens. Matter* **29**, 185901 (2017)] is 2.726 %, which is in similar level with previous report [*ACS Appl. Mater. Interfaces* **12**, 16350 (2020)]. We emphasize that we tried to minimize angle distortion and lattice strain to ensure the reliability of the calculated total energy.

3) The authors claimed/showed long cycle stability of the interface (800 cycles at room temperature) however there is no computational support for this point. To investigate the interactions between two surfaces, for example between Li and LLZTO, the *ab initio* molecular dynamics (AIMD) simulations at operating temperature are needed. Did the author think in this direction?

I do not agree with the answer. In our computational community, for Li metal anode system, at very least, we all usually can perform AIMD for the interface at $T=300$ K about 20 to 40 ps and hence, I would suggest the authors should perform this step to have a solid conclusions for their paper.

Author reply:

We appreciate the reviewer for suggesting interfacial AIMD simulations to support the long cycle stability. In our previous response, we agreed to the reviewer that the room temperature AIMD simulation would be helpful to directly understand the interfacial stability, but we also argued that the nanosecond-scale *ab initio* MD simulation of our trilayer model (LLZTO-Ag-AgC) would be computationally challenging.

We definitely understand that AIMD simulations of interfaces for tens of picoseconds is straightforward, but we believed that nanosecond-scale simulations should be required to comprehensively evaluate the interfacial stability. However, during a literature survey, we found out that AIMD simulations of interfaces for tens of picoseconds are sufficient to evaluate the preliminary interfacial reactivity or stability, as the reviewer suggested [*Chem. Mater.* **30**, 163 (2018), *ACS Appl. Mater. Interfaces* **13**, 43734 (2021), *Chem. Eng. J.* **434**, 134679 (2022), *Adv. Mater. Interfaces* **8**, 2100624 (2021)].

Therefore, we performed room-temperature (300 K) AIMD simulations for LLZTO/Li₉Ag₄ interface. Since LLZTO is in contact with sputtered Ag layer in our cell (Fig. 5a), we expect that the primary interface would consist of LLZTO and Li-Ag alloy during charge/discharge cycles. For AIMD simulations, we first stabilized the interface by slow heating (0 K to 300 K for 2 ps). After the equilibration, a production run of 40 ps was performed to observe if any interfacial reaction occurs. Supplementary Figure 8 shows the snapshots of LLZTO/Li₉Ag₄ interface structures during NVT AIMD simulations at 300 K. Clearly, any kind of chemical interaction or atomic interdiffusion was not observed between LLZTO and Li₉Ag₄ for 40 ps, indicating that LLZTO/Li₉Ag₄ interface is chemically stable. This observation computationally corroborates the long cycle stability of our lithium metal battery. We additionally performed AIMD simulations for LLZTO/Li interface (Figure R1), and confirmed that the interface is stable for at least 20 ps, as reported in previous computational works [*J. Electrochem. Soc.* **167**, 100537 (2020), *ACS Appl. Mater. Interfaces* **12**, 55510 (2020)].

Supplementary Figure 8. Structure evolution of LLZTO/Li₉Ag₄ interface upon NVT AIMD simulation.

Figure R1. Structure evolution of LLZTO/Li interface upon NVT AIMD simulation.

In response with the reviewer's comment, we revised the manuscript as follows:

Page 7

Original text

Structural relaxation of interfaces was conducted in two steps: distance between two surfaces was first optimized; then, one of the surface structures was laterally shifted to determine the most stable alignment between two surface structures. The adhesion energy (W_{ad}) was obtained by comparing the energies of interface structure and isolated surface structures.

$$W_{ad} = \frac{1}{A} (E_{interface} - E_{substrate} - E_{film}), \quad (1)$$

where A is the area of interface. $E_{interface}$, $E_{substrate}$, and E_{film} are calculated energies of interface, substrate, and film structures, respectively.

Revised text

Structural relaxation of interfaces was conducted in two steps: distance between two surfaces was first optimized; then, one of the surface structures was laterally shifted to determine the most stable alignment between two surface structures. The adhesion energy (W_{ad}) was obtained by comparing the energies of interface structure and isolated surface structures.

$$W_{ad} = \frac{1}{A}(E_{interface} - E_{substrate} - E_{film}), \quad (1)$$

where A is the area of interface. $E_{interface}$, $E_{substrate}$, and E_{film} are calculated energies of interface, substrate, and film structures, respectively.

Ab initio molecular dynamics (AIMD) simulations were performed to observe the structure evolution of interfaces. Starting from interface models generated by the method described above, we first stabilized the interface by slow heating (0 K to 300 K for 2 ps). After the equilibration, a production run of 40 ps was performed in NVT ensemble at 300 K.

Page 16

Original text

Nevertheless, our interlayer could successfully protect LLZTO from dendrite penetration, which helped realize a high volumetric energy density and an excellent long-term cycle stability for 800 cycles (Figure 5e).

Revised text

Nevertheless, our interlayer could successfully protect LLZTO from dendrite penetration, which helped realize a high volumetric energy density and an excellent long-term cycle stability for 800 cycles (Figure 5e). The outstanding cycle stability could be also attributed to the chemical stability of LLZTO/interlayer interface. In our AIMD simulations (Supplementary Figure 8), it is clearly shown that LLZTO/Li₉Ag₄ interface remains intact for at least 40 ps, which suggests reasonable chemical stability of the interface between LLZTO and Ag/AgC interlayer.

7) lines 226-254 the authors discussed and concluded about Li transport through the interlayer without any computational/theoretical evidence. The authors can use cNEB method to calculate Li migration barriers at the interface to support the observation.

Author reply:

We appreciate the reviewer for the valuable comment on Li transport kinetics through the interlayer. In our previous response, we discussed that the major Li transport mechanism through our mixed-conducting interlayer is creep, and suggested an activation energy value for Li creep (~0.38 eV), citing the work of LePage *et al.* [*J. Electrochem. Soc.* **166**, A89 (2019)]. As the reviewer suggested, we conducted NEB calculations to further discuss the Li transport kinetics through the AgC interlayer (lines 226-254 in the original manuscript, lines 221-230 in the 1st revised manuscript).

Assuming power-law creep, the strain rate $\dot{\epsilon}$ could be described by the equation

$$\dot{\epsilon} = A_c \sigma^m \exp\left(\frac{-Q_c}{RT}\right),$$

where σ is an uniaxial stress, A_c is a material-specific creep parameter, m is a creep exponent, R is a molar gas constant, T is a temperature and Q_c is an activation energy for dislocation climb [*J. Electrochem. Soc.* **166**, A89 (2019)]. In our cell configuration, Li is transported through non-graphitic carbon-based interlayer. Therefore, at the surface of carbon, activation energy (Q_c) could be different from that at the bulk Li. Following the reviewer's suggestion, we tried to estimate Q_c at the carbon interface by NEB calculation. As shown in Supplementary Figure 3, activation barrier of Li migration at the surface of carbon is 0.25 eV, which is smaller than the Q_c of bulk Li (~0.38 eV). This indicates the possibility of favorable Li transport kinetics in our carbon-based interlayer.

While the discussion in lines 226-254 (lines 221-230 in the 1st revised manuscript) is about the Li transport through the interlayer, Li transport kinetics between LLZTO and interlayer is also important. In order to evaluate the Li diffusion barriers, we performed NEB calculations for two possible Li diffusion pathways at the interface, as the reviewer suggested. According to our calculation, kinetically resolved activation barrier (KRA) of Li diffusion at LLZTO/Li₉Ag₄ interface is in the range of 0.15 ~ 0.3 eV, suggesting fast Li transport kinetics at the interface (Figure R2). Note that KRA is used due to the directional dependence of migration barrier [*J. Mater. Chem. A* **8**, 19965 (2020)], such as in the case of migration at the interface [*Adv. Theory Simul.* **2**, 1900028 (2019)]. This result is further supported by a recent experimental observation [*Adv. Energy Mater.* **10**, 1903993 (2020)], where low energy barrier of Li nucleation at Li-Ag alloy was demonstrated.

Supplementary Figure 3. (a) Schematics of Li distribution inside carbon-based interlayer. (b) Li diffusion pathways along the surface of carbon. (c) Energy profiles of Li diffusion along the surface of carbon.

Figure R2. (a) Calculated Li diffusion pathways at LLZTO/Li₉Ag₄ interface. (b) Energy profiles along with the Li diffusion.

In response with the reviewer's comment, we revised the manuscript for clarification as follows:

Page 7

Original text

Structural relaxation of interfaces was conducted in two steps: distance between two surfaces was first optimized; then, one of the surface structures was laterally shifted to determine the most stable alignment between two surface structures. The adhesion energy (W_{ad}) was obtained by comparing the energies of interface structure and isolated surface structures.

$$W_{ad} = \frac{1}{A}(E_{interface} - E_{substrate} - E_{film}), \quad (1)$$

where A is the area of interface. $E_{interface}$, $E_{substrate}$, and E_{film} are calculated energies of interface, substrate, and film structures, respectively.

Revised text

Structural relaxation of interfaces was conducted in two steps: distance between two surfaces was first optimized; then, one of the surface structures was laterally shifted to determine the most stable alignment between two surface structures. The adhesion energy (W_{ad}) was obtained by comparing the energies of interface structure and isolated surface structures.

$$W_{ad} = \frac{1}{A}(E_{interface} - E_{substrate} - E_{film}), \quad (1)$$

where A is the area of interface. $E_{interface}$, $E_{substrate}$, and E_{film} are calculated energies of interface, substrate, and film structures, respectively.

Nudged elastic band (NEB) calculations³¹ were conducted to evaluate the energy barrier of Li diffusion at the surface of carbon. Seven intermediate images were generated along the diffusion pathways, and structure relaxation was performed to determine the energy profile.

31. Henkelman, G., Uberuaga, B. P. & Jónsson, H. A climbing image nudged elastic band method for finding saddle points and minimum energy paths. *J. Chem. Phys.* **113**, 9901–9904 (2000).

Page 9

Original text

If the time for the reduced Li to move toward the current collector through the interlayer via creep is insufficient, Li would be undesirably plated near the solid electrolyte, although it is thermodynamically unfavorable. In other words, Li transport through the interlayer should be faster than Li influx from the cathode side, which is dependent on the charge rate.

Revised text

If the time for the reduced Li to move toward the current collector through the interlayer via creep is insufficient, Li would be undesirably plated near the solid electrolyte, although it is thermodynamically unfavorable. In other words, Li transport through the interlayer should be faster than Li influx from the cathode side, which is dependent on the charge rate. In this regard, carbon-based interlayer could aid Li transport. According to our NEB calculations, activation barrier of Li diffusion along the surface of carbon is as low as 0.25 eV (Supplementary Figure 3), which suggests the possibility of fast Li diffusion through the carbon-based interlayer.

REVIEWERS' COMMENTS

Reviewer #2 (Remarks to the Author):

The authors answer my concern questions, I would suggest that the paper should be published in Nature Com.